# InfoGeo: Information-Theoretic Object-Centric Learning for Cross-View Generalizable UAV Geo-Localization

Hongyang Zhang [* 1]  Maonan Wang [* 2 3]  Ziyao Wang [1]  Hongrui Yin [1]  Man On Pun [1]

## Abstract

Cross-view geo-localization (CVGL) is fundamental for precise localization and navigation in GPS-denied environments, aiming to match ground or UAV imagery with satellite views. Existing approaches often rely on global feature alignment, but they suffer from substantial domain shifts induced by varying regional textures and weather conditions. This issue becomes even more pronounced in UAV-based scenarios, where the broader perspective inevitably introduces dense, fine-grained objects, creating significant visual clutter. To address this, we draw inspiration from Object-Centric Learning (OCL) and propose InfoGeo, an information-theoretic framework designed to enhance robustness and generalization. InfoGeo reformulates the optimization as an information bottleneck process with two core objectives: (i) maximizing view-invariant information by aligning the object-centric structural relations across views, and (ii) minimizing view-specific noisy signals through cross-view knowledge constraints. Extensive evaluations across diverse benchmarks and challenging scenarios demonstrate that InfoGeo significantly outperforms state-of-the-art methods.

## 1. Introduction

Cross-view geo-localization (CVGL) aims to identify the same geographic location from images captured under different viewpoints, serving as a fundamental capability for various practical applications such as city management (Workman et al., 2015), robot navigation (Sarlin et al., 2019), and disaster monitoring (Li et al., 2026). To bridge the geometric gap between views, existing methods (Deuser et al., 2023; Zhou et al., 2025) mainly propose global feature alignment via contrastive learning, encouraging images to cluster closely in a high-level semantic space. Recently, UAV-based CVGL (Zheng et al., 2020) has emerged as a powerful paradigm, providing rich multi-view observations that facilitate more comprehensive scene understanding. However, current approaches face significant performance degradation when confronted with domain shifts induced by varying geographic regions and weather conditions. This limitation stems from the reliance of global-based methods on coarse discrimination, which overlooks fine-grained object-level correspondences. Consequently, their scalability is severely restricted in UAV scenarios featuring dense object distributions and complex spatial layouts.

To address the above limitations, we draw inspiration from human visual cognition for object-centric reasoning (Kejriwal et al., 2024) to establish cross-view consensus through object-level comparative analysis. Recently, Object-Centric Learning (OCL) (Locatello et al., 2020; Fan et al., 2024) has emerged as a promising paradigm for scene understanding, enabling constructing object-centric concepts and binding view-shared object-level relations. Therefore, OCL provides a principled foundation for CVGL by facilitating the extraction of view-invariant information to boost robustness and generalization. Although some OCL-based works (Li et al., 2020; Zhao et al., 2023) have targeted multi-view scenes, they mainly focus on environments with uncluttered backgrounds and sparse, salient objects. However, UAV-based CVGL involves visual clutter caused by dense object distributions, alongside cross-view spatial discrepancies and seasonal changes. Consequently, directly applying these OCL approaches yields suboptimal performance in such unstructured scenarios.

To tackle these challenges, we extract robust representations by focusing on intrinsic geometric structures while filtering out environmental interference. As illustrated in Figure 1 (a), visual content across different views can be decomposed into two distinct components: 1) view-invariant information (highlighted in orange), which represents the stable structural semantics essential for matching, and 2) view-specific nuisances (highlighted in green), such as dynamic objects

---

[*]Equal contribution [1]School of Science and Engineering, The Chinese University of Hong Kong (Shenzhen), Guangdong 518172, China [2]Department of Mechanical and Automation Engineering, The Chinese University of Hong Kong, Hong Kong SAR 999077, China [3]Shanghai AI Laboratory, Shanghai 200032, China. Correspondence to: Man On Pun <simonpun@cuhk.edu.cn>.

*Proceedings of the 43rd International Conference on Machine Learning*, Seoul, South Korea. PMLR 306, 2026. Copyright 2026 by the author(s).

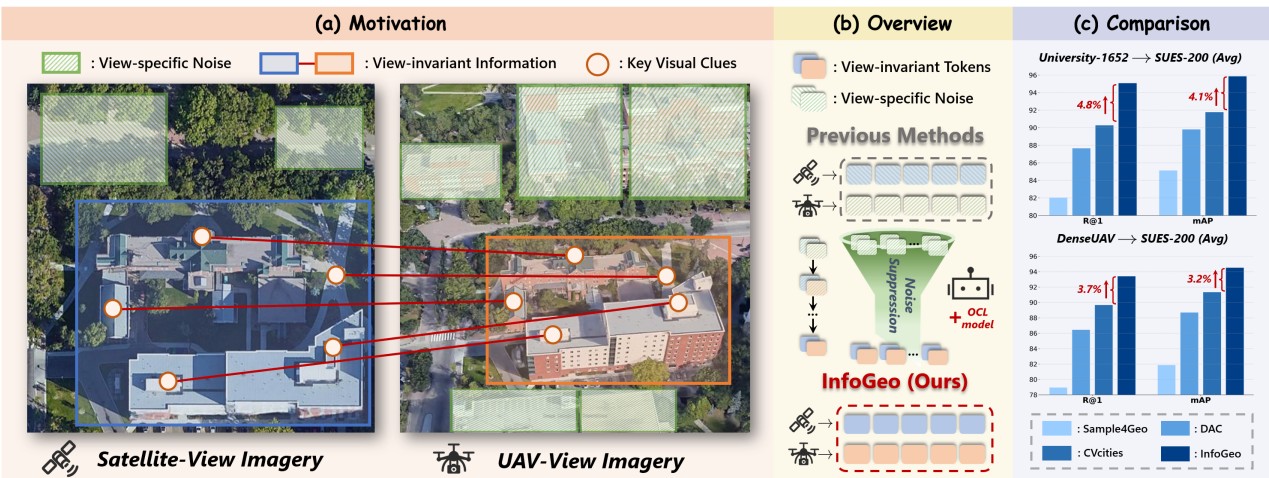

*Figure 1.* (a) The illustration of our motivation. Cross-view images can be decomposed into the *view-invariant information* and *view-specific noise*, while paired data can be matched through key visual clues. (b) The overview of *cross-view object-centric learning* process, the main target is to extract view-invariant tokens by compressing the view-specific noise. (c) Comparison with recent state-of-the-art methods on the two transfer scenarios, our method is built upon the model *CVcities* and achieves superior generalization ability.

and occlusion. Based on this insight, we introduce InfoGeo, a cross-view object-centric framework designed to explicitly extract these view-invariant concepts for more generalizable representation learning. In Figure 1 (b), our approach not only aligns these consistent concepts but also suppresses view-specific noise. This design principle creates a unified optimization objective grounded in the Information Bottleneck (IB) theory (Tishby et al., 1999; Li et al., 2025), which guides the model to maximize relevant information while compressing irrelevant signals.

Specifically, our method implements this vision through a novel Cross-View Visual Concept Reasoner equipped with an Object-Centric Visual Augmentation (OCVA) module. The core of our reasoning process consists of two synergistic pathways within the Cross-View OCL framework: Cross-View Adaptive Concept Selection (CACS) and Concept Structural Relational Reasoning (CSRR). On one hand, CACS functions as a selective filter, leveraging concept-level experts to distill view-shared information while suppressing view-specific nuisances. On the other hand, CSRR harnesses concept graph relations to maintain structural consistency across viewpoints. Following this reasoning phase, the OCVA module seamlessly fuses the object-centric features into global high-level semantics via feature-wise linear modulation. As demonstrated in Figure 1 (c), this design leads to superior generalization performance over existing CVGL approaches. Overall, our key contributions can be summarized as follows:

- Our work is the first to explore CVGL with object-centric learning. Unlike holistic approaches, InfoGeo explicitly disentangles view-invariant geometric entities from environmental clutter, offering a robust solution to severe domain shifts in unseen regions.

- We reformulate the cross-view matching problem via the Information Bottleneck principle. This theoretical optimization guides our framework to simultaneously maximize structural consensus and compress view-specific nuisances, ensuring efficient and noise-resilient representation learning.

- We conduct extensive evaluations on four UAV benchmarks. InfoGeo achieves new state-of-the-art generalization performance, demonstrating superior robustness across cross-region and multi-weather scenarios.

The remainder of this paper is organized as follows: Section 2 provides the preliminaries. Section 3 and Section 4 detail the theoretical foundation and the proposed architecture, respectively. We present extensive experiments in Section 5 and discuss related works of CVGL and Multi-View OCL in Appendix Section A.

## 2. Preliminaries

### 2.1. Problem Formulation

We investigate the task of generalizable CVGL under cross-region distribution shifts. Let $\mathbf{X}^q$ and $\mathbf{X}^g$ denote the query and gallery images, respectively. Each cross-view pair $(x^q, x^g)$ is captured at the geographic location $y \in \mathbf{Y}$. We aim to train a model $f_\theta$ with trainable parameters $\theta$ to extract generalizable visual representations. Specifically, $f_\theta$ takes image pairs $(\mathbf{X}^q, \mathbf{X}^g)$ from source regions and produces the scene-level descriptors $(\mathbf{F}^q, \mathbf{F}^g)$ for localization, while evaluation is conducted on unseen domains.

### 2.2. Object-Centric Learning

Object-Centric Learning with Slot Attention (Locatello et al., 2020) compresses dense visual observations into

sparse latent slots, each capturing an underlying concept. We follow the design of slot attention module in (Zhao et al., 2025) and apply it to our framework. The overview is illustrated in Figure 2.

Cross-view images $\mathbf{X}^{(v)} \in \mathbb{R}^{H \times W \times C}$ in view $v \in \{q, g\}$ are fed into a frozen visual encoder to obtain feature maps $\mathbf{Z}^{(v)} \in \mathbb{R}^{\frac{H}{14} \times \frac{W}{14} \times C}$. Then, the slot encoder $\phi_e$ extracts the initial slot vectors $\mathbf{S}_0^{(v)} \in \mathbb{R}^{K \times C}$ from $\mathbf{Z}^{(v)}$, where each of the $K$ slots corresponds to a distinct concept. The aggregator $\phi_a$ is further employed as an iterative refinement process over $T$ iterations to project the features onto $K$ slots with the aggregation attention map $\mathbf{A}_a^{(v)}$:

$$\mathbf{S}_i^{(v)}, \mathbf{A}_a^{(v)}(i) = \phi_a(\mathbf{S}_{i-1}^{(v)}, \mathbf{Z}^{(v)}), \quad i = 1, \dots, T$$
$$\mathbf{S}^{(v)} \equiv \mathbf{S}_T^{(v)}. \tag{1}$$

Subsequently, the aggregated slots $\mathbf{S}^{(v)}$ are decoded by an auto-regressive based decoder $\phi_d$ to reconstruct the inputs and obtain $\mathbf{Z}'^{(v)} \in \mathbb{R}^{H' \times W' \times C'}$, with some clues $\mathbf{C}^{(v)}$:

$$\mathbf{Z}'^{(v)}, \mathbf{A}_d^{(v)} = \phi_d(\mathbf{C}^{(v)}, \mathbf{S}^{(v)}), \tag{2}$$

where $\mathbf{A}_d^{(v)} \in \mathbb{R}^{K \times H' \times W'}$ is the decoding attention map, which is further used in the cross-view OCL module. $\mathbf{Z}'^{(v)}$ denotes the reconstruction of $\mathbf{Z}^{(v)}$, while $\mathbf{C}^{(v)}$ is instantiated as $\mathbf{Z}^{(v)}$ in this work.

The supervision signal of OCL comes from the slot reconstruction loss via the Mean Squared Error (MSE):

$$\mathcal{L}_{\text{rec}} = \sum_{v \in \{q, g\}} \text{MSE}(\mathbf{Z}'^{(v)}, \mathbf{Z}^{(v)}). \tag{3}$$

## 3. Information Bottleneck Theory for CVGL

Although different views of the same location $\mathbf{Y}$ exhibit large gaps, they share consistent semantic structures. The feature maps $\mathbf{Z}^{(v)}$ are extracted from images $\mathbf{X}^{(v)}$ using a frozen visual encoder, and consist of both task-relevant information and view-specific noise. Our task aims to extract compact representations $\hat{\mathbf{Z}}^{(v)}$ that retain task-relevant features while suppressing view-specific noise. Accordingly, we adopt the Information Bottleneck (IB) principle (Tishby et al., 1999) to formalize our task.

**Theorem 3.1** (View-Invariant Information Bottleneck Principle). *The representations $\hat{\mathbf{Z}}^{(v)}$ are optimal for CVGL when they maximize mutual information (MI) with the geographic identity $\mathbf{Y}$ while minimizing task-irrelevant information inherited from the original features $\mathbf{Z}^{(v)}$:*

$$\max_{v \in \{q, g\}} \mathcal{L}_{IB}^{(v)} = I(\hat{\mathbf{Z}}^{(v)}; \mathbf{Y}) - \beta I(\hat{\mathbf{Z}}^{(v)}; \mathbf{Z}^{(v)} \mid \mathbf{Y}), \tag{4}$$

*where $\beta \geq 0$ regulates the trade-off between preserving task-relevant information and compressing task-irrelevant information.*

**Theorem 3.2** (Cross-View Mutual Information Bound). *In the context of cross-view learning, models are trained with one-to-one image pairs to capture spatial correspondences without explicit class-level supervision (Li et al., 2024a). Under this assumption, we constrain the first term in information bottleneck flow via the upper bound $I(\hat{\mathbf{Z}}^q; \hat{\mathbf{Z}}^g)$:*

$$I(\hat{\mathbf{Z}}^q; \mathbf{Y}) \leq I(\hat{\mathbf{Z}}^q; \hat{\mathbf{Z}}^g), \quad I(\hat{\mathbf{Z}}^g; \mathbf{Y}) \leq I(\hat{\mathbf{Z}}^q; \hat{\mathbf{Z}}^g). \tag{5}$$

These inequalities follow the Markov chain relations and the data processing inequality (Beaudry & Renner, 2012) (more proof is provided in Appendix Section B.2).

Maximizing $I(\hat{\mathbf{Z}}^q; \hat{\mathbf{Z}}^g)$ encourages the preservation of view-shared relationships. In practice, we optimize this objective with a variational contrastive lower bound (e.g., InfoNCE (Van den Oord et al., 2018)), which provides low-variance estimation and good scalability.

**Proposition 3.3** (Cross-View Information Bottleneck Theory). *Under the conditional assumption that $\hat{\mathbf{Z}}^q \perp \hat{\mathbf{Z}}^g \mid \mathbf{Y}$ (see Appendix Section B.2), the gallery representation $\hat{\mathbf{Z}}^g$ can be regarded as a compact proxy for the location identity $\mathbf{Y}$. Since all view-shared information originates from $\mathbf{Y}$, $I(\hat{\mathbf{Z}}^q; \hat{\mathbf{Z}}^g)$ provides an upper-bound-driven proxy for $I(\hat{\mathbf{Z}}^q; \mathbf{Y})$ (Theorem 3.2), under the constraint that $\hat{\mathbf{Z}}^g$ preserves task-relevant semantics while suppressing view-specific nuisances.*

*Meanwhile, the remaining information in $\mathbf{Z}^q$ that cannot be inferred from $\hat{\mathbf{Z}}^g$ is predominantly view-specific and task-irrelevant, motivating a view-specific compression term. Accordingly, the information bottleneck objective $I(\hat{\mathbf{Z}}^q; \mathbf{Z}^q \mid \mathbf{Y})$ can be approximated by $I(\hat{\mathbf{Z}}^q; \mathbf{Z}^q \mid \hat{\mathbf{Z}}^g)$, thereby linking cross-view mutual information maximization with principled IB-based representation learning.*

*Combining these observations, the Information Bottleneck objective $\mathcal{L}_{\text{IB}}^{q \to g}$ for CVGL can be formulated as:*

$$\max \mathcal{L}_{\text{IB}}^{q \to g} = I(\hat{\mathbf{Z}}^q; \hat{\mathbf{Z}}^g) - \beta I(\hat{\mathbf{Z}}^q; \mathbf{Z}^q \mid \hat{\mathbf{Z}}^g), \tag{6}$$

*where $\hat{\mathbf{Z}}^g$ acts as an implicit sufficient statistic of the identity geographic variable $\mathbf{Y}$. By symmetry, the gallery-to-query objective $\mathcal{L}_{\text{IB}}^{g \to q}$ is defined analogously.*

This proposition provides the information-theoretical foundation for the optimization of cross-view OCL.

## 4. Cross-view Visual Concept Reasoner

The overview of InfoGeo is illustrated in Figure 2, where Cross-view Visual Concept Reasoner is a plug-and-play network. In the network, we first introduce Cross-View Object-Centric Learning to obtain robust representations in the perspective of the Information Bottleneck principle (Section 4.1). In Section 4.2, the OCVA module is proposed

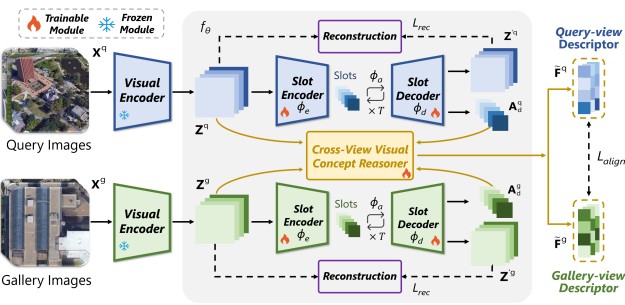

*Figure 2.* The overview of our proposed framework InfoGeo.

to incorporate object-centric representations into the scene-level descriptors, enabling fine-grained discrimination with view-invariant semantics. The detailed information of them is illustrated in Figure 3. Lastly, the overall training objective is given in Section 4.3.

### 4.1. Cross-View Object-Centric Learning

We integrate IB theory into Cross-View Object-Centric Learning by introducing Cross-view Adaptive Concept Selection (CACS) to effectively suppress view-specific noise, while Concept Structural Relational Reasoning (CSRR) further preserves view-invariant relational structures.

#### 4.1.1. CROSS-VIEW ADAPTIVE CONCEPT SELECTION

We propose a CACS module to explore the view-shared concepts through cross-view attention modules. Let $\mathbf{A}_d^q \in \mathbb{R}^{K \times H \times W}$ be the decoding attention maps in the query view and corresponding $\mathbf{A}_d^g$ in the gallery view. We employ a multi-head attention (MHA) for cross-view feature fusion:

$$\mathbf{A}_d^{q \to g} = \text{LayerNorm}(\mathbf{A}_d^q + \text{MHA}(Q, K, V)),$$
$$\text{where } Q = \mathbf{A}_d^q, \quad K = \mathbf{A}_d^g, \quad V = \mathbf{A}_d^g. \quad (7)$$

However, occlusions and illumination variations would lead to the discrepancy of concept-level information across viewpoints. To address this, we introduce an adaptive expert concept router $\psi(\cdot)$ that dynamically reweights concept-level decoding attention maps according to their cross-view relevance. Given the fused attention maps, the refined concept representations are formulated as:

$$\hat{\mathbf{A}}_d^{(v)} = \mathbf{W}_{\text{cv}} \odot \mathbf{A}_d^{(v)}, \text{ where } \mathbf{W}_{\text{cv}} = \psi(\mathbf{A}_d^{(v) \to (\bar{v})}). \quad (8)$$

where $\psi(\cdot)$ is a learnable linear layer, $\mathbf{W}_{\text{cv}} \in \mathbb{R}^{K \times 1}$ denotes the adaptive weights, and $\odot$ is the Hadamard product.

To explicitly encourage concept routers to suppress task-irrelevant noise in each view, we further impose a regularization constraint on $\mathbf{W}_{\text{cv}}$ that promotes decisive selection and inter-concept diversity. Specifically, we define the following objective:

$$\mathcal{L}_{\text{cacs}} = \mathbb{E}\left[\mathbf{W}_{\text{cv}} \odot (1 - \mathbf{W}_{\text{cv}})\right] - \mathbb{E}\left[\text{Var}(\mathbf{W}_{\text{cv}})\right], \quad (9)$$

where the first term penalizes ambiguous weights to encourage confident and near-binary selection, the second term promotes variance among concepts, preventing the routers from assigning similar responses to all concepts and encouraging diverse concept activation.

Consequently, the module highlights view-shared concepts that are consistently observed across views and mitigates the impacts of view-specific artifacts.

#### 4.1.2. CONCEPT STRUCTURAL RELATIONAL REASONING

We construct concept graphs and align their relations through structural relational reasoning to tackle severe spatial discrepancies in concepts across different views.

To capture high-order dependencies among semantic concepts, we formally model the concept graphs $\mathbf{G}^{(v)} \in \mathbb{R}^{K \times K}$ through $\hat{\mathbf{A}}_d^{(v)}$, with nodes as concepts and edges encoding pairwise semantic similarities through cosine distance. And Laplacian Eigenmaps (Belkin & Niyogi, 2003) are applied to preserve local geometry and semantic proximity, yielding a view-robust spectral embedding that enables consistent and reliable concept alignment across views. The normalized graph Laplacian $\mathbf{L}^{(v)}$ is then constructed to enforce graph smoothness:

$$\mathbf{L}^{(v)} = \mathbf{I} - \left(\mathbf{D}^{(v)}\right)^{-\frac{1}{2}} \mathbf{G}^{(v)} \left(\mathbf{D}^{(v)}\right)^{-\frac{1}{2}}, \quad (10)$$

where $\mathbf{D}^{(v)}$ is the degree matrix of $\mathbf{G}^{(v)}$. Then, we derive the structural relations through spectral decomposition:

$$\mathbf{L}^{(v)} = \mathbf{V}^{(v)} \mathbf{\Lambda}^{(v)} \left(\mathbf{V}^{(v)}\right)^{\top}, \quad (11)$$

where $\mathbf{V}^{(v)}$ and $\mathbf{\Lambda}^{(v)}$ represent the eigenvectors and eigenvalues, respectively, capturing the intrinsic structural relations of the concept graph.

Furthermore, we discard the trivial eigenvectors associated with zero eigenvalues and retain the top $r$ non-trivial eigenvectors ($r = 4$), following standard practice in spectral embedding (Von Luxburg, 2007), to obtain the most important representations $\mathbf{U}^{(v)} \in \mathbb{R}^{K \times r}$:

$$\mathbf{U}^{(v)} = \text{Top}^r(\mathbf{V}^{(v)}) = \left[\mathbf{v}_1^{(v)}, \dots, \mathbf{v}_r^{(v)}\right], \quad (12)$$

which preserves the intrinsic graph topology while remaining robust to significant cross-view appearance variations.

While the spectral embeddings $\mathbf{U}^q$ and $\mathbf{U}^g$ are derived independently, their representations are not directly comparable due to inherent orthogonal ambiguities in the underlying spectral subspaces (Umeyama, 2002). To enforce cross-view structural consistency, we align the two embeddings

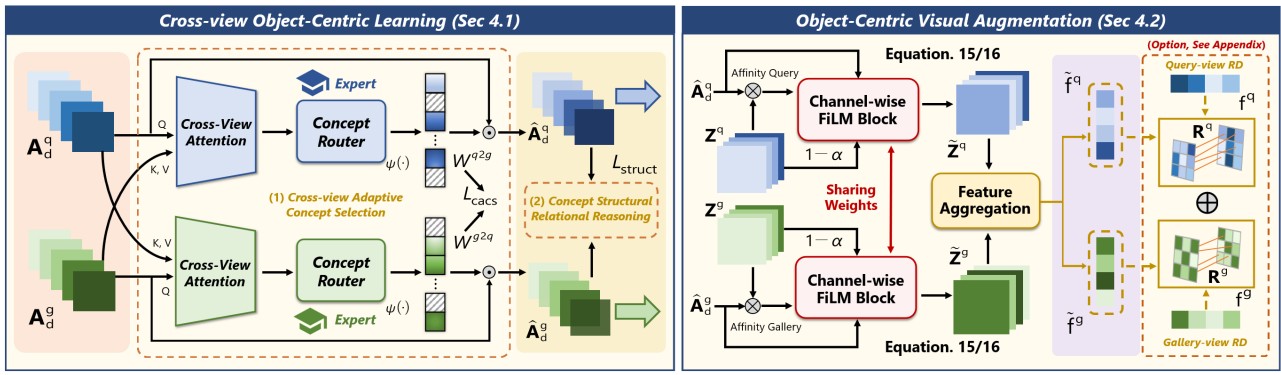

*Figure 3.* The training pipeline of Cross-view Visual Concept Reasoner, which pioneers an IB theory based framework for cross-view OCL through two synergistic components: 1) Cross-view Adaptive Concept Selection, and 2) Concept Structural Relational Reasoning. Object-Centric Visual Augmentation is further proposed to integrate object-centric representations into the global scene-level descriptors.

via the orthogonal procrustes loss $\mathcal{L}_{\text{struct}}$:

$$\mathcal{L}_{\text{struct}} = \min_{\mathbf{Q} \in \mathcal{O}(r)} \|\mathbf{U}^q - \mathbf{U}^g \mathbf{Q}\|_2^2, \quad (13)$$

where $\|\cdot\|_2^2$ denotes the $L_2$ norm and $\mathcal{O}(r) = \{\mathbf{Q} \in \mathbb{R}^{r \times r} \mid \mathbf{Q}^\top \mathbf{Q} = \mathbf{I}\}$ is the orthogonal group (reflecting the inherent rotational ambiguity in spectral representations).

Overall, this reasoning process resolves cross-view concept ambiguity and enables robust relational reasoning via concept graph matching in a view-shared space.

## 4.2. Object-Centric Visual Augmentation

To inject object-centric representations from Section 4.1 into the global representations, we propose Object-Centric Visual Augmentation (OCVA) module in this section.

Specifically, we employ two channel-wise FiLM blocks following (Perez et al., 2018) on both branches to incorporate object-centric representations into global representations. For each view $v$, let $\mathbf{Z}^{(v)} \in \mathbb{R}^{C \times H \times W}$ denote the global feature map and $\hat{\mathbf{A}}_d^{(v)} \in \mathbb{R}^{K \times H \times W}$ represent the $K$ concept-level attention maps. To effectively capture the object-centric semantics, we first compute the affinity vector $\mathbf{v}_h^{(v)}$ between the $k$-th attention map $\hat{\mathbf{A}}_d^{(v)}(k)$ and $\mathbf{Z}^{(v)}$:

$$\mathbf{v}_h^{(v)} = \mathcal{G}(\sum_{k=1}^{K} \sum_{i=1}^{HW} \hat{\mathbf{A}}_d^{(v)}(k) \cdot \mathbf{Z}_i^{(v)}), \quad (14)$$

where $\mathcal{G}(\cdot)$ denotes a learnable transformation that projects the affinities into a unified semantic space.

Subsequently, we derive a channel-wise modulation vector $\gamma^{(v)}$ to adaptively calibrate the global representation $\mathbf{Z}^{(v)}$. The modulated feature map $\hat{\mathbf{Z}}^{(v)}$ is formulated as:

$$\hat{\mathbf{Z}}^{(v)} = \mathbf{Z}^{(v)} \odot \left(1 + \gamma^{(v)}\right), \text{ where } \gamma^{(v)} = \sigma\left(\mathcal{P}(\mathbf{v}_h^{(v)})\right), \quad (15)$$

where $\mathcal{P}(\cdot)$ denotes a learnable projection that maps the

affinity vector into the feature channel space, $\sigma(\cdot)$ represents the sigmoid activation function.

This process enables object-centric feature modulation by adaptively emphasizing channels that are more semantically aligned with the learned concepts. The modulated features are subsequently fused with the global features to obtain the output feature maps $\tilde{\mathbf{Z}}^{(v)}$:

$$\tilde{\mathbf{Z}}^{(v)} = \alpha \hat{\mathbf{Z}}^{(v)} + (1 - \alpha)\mathbf{Z}^{(v)}, \quad (16)$$

where $\alpha$ is the balancing factor in the module.

Finally, the corresponding scene-level descriptors $\tilde{\mathbf{F}}^{(v)}$ are obtained via a view-shared feature aggregator $\text{Agg}(\cdot)$:

$$\tilde{\mathbf{F}}^{(v)} = \text{Agg}(\tilde{\mathbf{Z}}^{(v)}). \quad (17)$$

The above descriptors can be directly used for retrieval. To improve inference efficiency, we also propose a relational distillation (RD) module (Park et al., 2019) to transfer object-centric instance-level relations from $\tilde{\mathbf{F}}^{(v)}$ to the vanilla descriptors $\mathbf{F}^{(v)}$ (obtained via $\text{Agg}(\cdot)$). By optimizing $\mathcal{L}_{\text{distill}}$, RD enables the vanilla descriptors to inherit object-centric relational knowledge. The details are presented in Appendix Section B.4.

## 4.3. Applying IB theory to generalizable CVGL

### 4.3.1. CROSS-VIEW OCL WITH IB THEORY.

We reformulate the optimization objective of Cross-View OCL by adapting the original upper and lower bounds of IB theory (see Theorem 3.3) to our scenario. The detailed derivation is deferred to Appendix Section B.2. Accordingly, Equation (6) can be further refined as:

$$\max \mathcal{L}_{\text{IB}} = \underbrace{I(\mathbf{U}^g; \mathbf{U}^q)}_{\mathcal{L}_{\text{struct}}} \underbrace{- [I(\hat{\mathbf{A}}_d^q; \mathbf{A}_d^q \mid \hat{\mathbf{A}}_d^g) + I(\hat{\mathbf{A}}_d^g; \mathbf{A}_d^g \mid \hat{\mathbf{A}}_d^q)]}_{\mathcal{L}_{\text{cacs}}}, \quad (18)$$

where $\mathbf{A}_d^{(v)}$ serves as the input variable (replacing $\mathbf{Z}^{(v)}$ in Equation (6)), while $\mathbf{U}^{(v)}$ and $\hat{\mathbf{A}}_d^{(v)}$ are both derived

*Table 1.* Comparisons with state-of-the-art models trained on University-1652 and DenseUAV datasets. The results on SUES-200 are reported as the average performance (%) across all subsets. (* method that uses relational distillation.)

| Model | University→SUES | | University → DenseUAV | | DenseUAV→SUES | | DenseUAV→University | |
|---|---|---|---|---|---|---|---|---|
| | R@1↑ | AP↑ | R@1↑ | AP↑ | R@1↑ | AP↑ | R@1↑ | AP↑ |
| MCCG (Shen et al., 2023) | 70.31 | 74.58 | 17.54 | 13.01 | 80.09 | 83.18 | 50.52 | 55.35 |
| Sample4Geo (Deuser et al., 2023) | 82.03 | 85.12 | 33.46 | 22.28 | 78.94 | 81.86 | 37.90 | 42.54 |
| DAC (Xia et al., 2024) | 87.65 | 89.80 | 37.81 | 26.99 | 86.44 | 88.70 | 45.41 | 50.03 |
| CAMP (Wu et al., 2024) | 88.34 | 90.49 | 36.94 | 26.39 | 86.79 | 88.93 | 44.11 | 48.60 |
| MFRGN (Wang et al., 2024) | 78.28 | 81.75 | 23.60 | 16.08 | 67.22 | 71.14 | 37.95 | 41.73 |
| EM-CVGL (Li et al., 2024b) | 74.45 | 78.51 | 13.47 | 10.19 | 56.43 | 61.54 | 32.76 | 38.05 |
| CVcities (Huang et al., 2024a) | 90.27 | 91.77 | 38.87 | 28.44 | 89.67 | 91.34 | 66.08 | 70.01 |
| Game4Loc (Ji et al., 2025a) | 86.75 | 89.07 | 17.76 | 13.62 | 67.19 | 71.40 | 26.47 | 31.05 |
| MMGeo (Ji et al., 2025b) | 85.01 | 87.60 | 22.88 | 18.54 | 79.05 | 82.05 | 33.92 | 38.28 |
| InfoGeo (Ours **w/o RD**) | **93.38**$_{+3.11}$ | **94.35**$_{+2.58}$ | **39.88**$_{+1.01}$ | **29.13**$_{+0.69}$ | **93.42**$_{+3.75}$ | **94.55**$_{+3.21}$ | **68.16**$_{+2.08}$ | **72.15**$_{+2.11}$ |
| InfoGeo*(Ours **w/ RD**) | **95.07**$_{+4.80}$ | **95.85**$_{+4.08}$ | **42.30**$_{+3.43}$ | **31.78**$_{+3.34}$ | **93.41**$_{+3.74}$ | **94.52**$_{+3.18}$ | **69.37**$_{+3.29}$ | **73.21**$_{+3.20}$ |

from $\hat{\mathbf{Z}}^{(v)}$ through the Markov chain information flow. This objective can be constructed from two aspects:

**Cross-View MI Maximization:** The first term encourages the model to preserve view-invariant information across different viewpoints. Specifically, minimizing $\mathcal{L}_{\text{struct}}$ corresponds to maximizing a variational lower bound on the mutual information at the latent concept level.

**View-Specific Noise Compression:** The second term aims to filter out task-irrelevant noise. CACS module reweights the attention maps of each slot concept through $\mathcal{L}_{\text{cacs}}$ for robust and generalizable representation learning.

The Cross-View OCL objective $\mathcal{L}_{\text{info}}$ can be defined as:

$$\mathcal{L}_{\text{info}} = \mathcal{L}_{\text{cacs}} + \mathcal{L}_{\text{struct}}. \tag{19}$$

### 4.3.2. OVERALL OPTIMIZATION

During the training process, we first learn object-centric representations via the slot reconstruction loss $\mathcal{L}_{\text{rec}}$ and then leverage $\mathcal{L}_{\text{info}}$ to regulate the information-theoretic based cross-view object-centric learning flow. Following previous CVGL works (Deuser et al., 2023; Huang et al., 2024a), we apply InfoNCE loss $\mathcal{L}_{\text{align}}$ to align the scene-level descriptors across different views, while optionally incorporating the additional relational distillation objective $\mathcal{L}_{\text{distill}}$ to reduce computation costs. The overall optimization can be calculated as follows:

$$\mathcal{L} = \mathcal{L}_{\text{align}} + \lambda_1 \mathcal{L}_{\text{rec}} + \lambda_2 \mathcal{L}_{\text{info}} + \lambda_3 \mathcal{L}_{\text{distill}}, \tag{20}$$

where $\lambda_1$, $\lambda_2$ and $\lambda_3$ are the trade-off parameters.

## 5. Experiment

### 5.1. Dataset and Evaluation Protocol

**Dataset.** In our evaluation, we conduct experiments on four UAV benchmarks: University-1652 (Zheng et al., 2020)

(Continuous sampling), SUES-200 (Zhu et al., 2023a) (Discrete sampling with flight altitudes ranging from 150 m to 300 m), DenseUAV (Dai et al., 2023) (Low-altitude and self-localization benchmark), and GTA-V (Ji et al., 2025a) (a challenging virtual UAV benchmark). Detailed dataset statistics are provided in Appendix Table 8.

**Evaluation Protocol.** Following previous works (Zheng et al., 2020; Ji et al., 2025a), we evaluate the retrieval results using Recall@K (R@K) and Average Precision (AP). A drone query is considered correctly localized if its ground-truth satellite image appears in the Top-K results. To assess spatial localization in GTA-V, we use the Spatial Distance Metric (SDM@3) for retrieval accuracy and report the meter-level distance between the top-1 retrieved results and ground-truth drone coordinates (Dis@1 / meter) as an intuitive measure of positioning.

### 5.2. Implementation Details

We adopt CVcities as the baseline where DINOv2-Base (Oquab et al., 2024) is the image encoder, while Mixer (Ali-Bey et al., 2023) is used as the feature aggregator to extract the scene-level descriptors. The image size is set to $448 \times 448$. The network is trained for 40 epochs with a batch size of 8 and conducted on a single 80GB NVIDIA A100 GPU. The model is optimized using AdamW with an initial learning rate of $6.5 \times 10^{-4}$. In OCL modules, the slot number $K$ is 16, $\alpha$ is set to 0.8. Both $\lambda_1$ and $\lambda_2$ are 0.05, while $\lambda_3$ is 0.85. All experiments are conducted on Drone→Satellite (Localization) with cross-domain evaluation, and only positive strategies are used in GTA-V dataset. Additional network structures and parameter settings are provided in Appendix Section C.1.

### 5.3. Comparing with State-of-the-art Models

**Cross-view Image retrieval.** As shown in Table 1 and Table 2, our method achieves the best overall performance

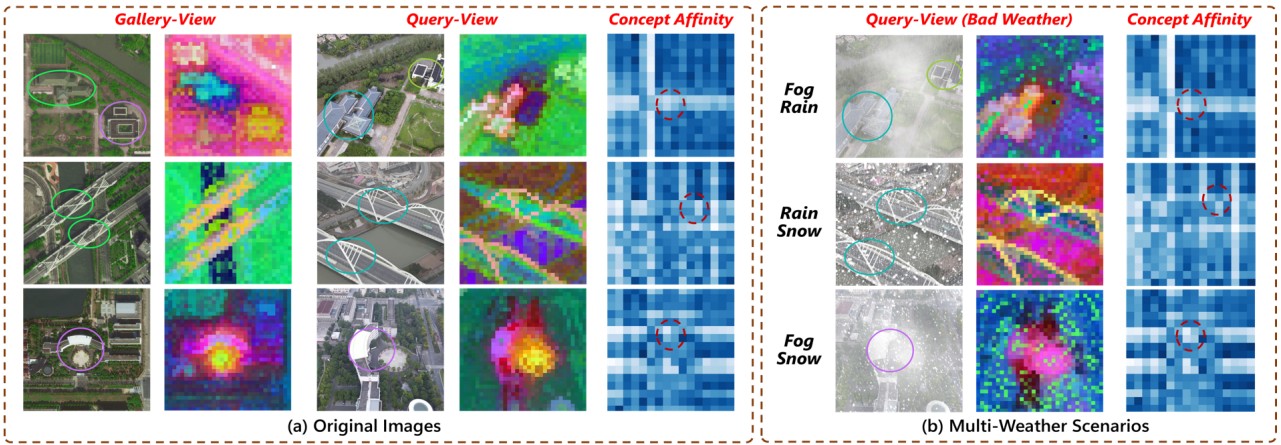

*Figure 4.* The PCA visualization and concept affinities of Object-Centric Representations $\hat{\mathbf{Z}}$. RGB values correspond to principal components. Circles are the view-shared landmarks. Concept affinities are calculated by the spatial-level cosine distance across viewpoints (darker colors indicate higher spatial similarity), while dashed circles highlight the feature-space regions that exhibit robustness.

*Table 2.* Performances (%) Comparisons with the state-of-the-art models on GTA-V (cross-area) dataset.

| Model | GTA-V (cross-area) | | | |
|---|---|---|---|---|
| | R@1↑ | AP↑ | SDM@3↑ | Dis@1↓ |
| Sample4Geo | 28.60 | 38.94 | 51.78 | 1078.33 |
| DAC | 53.31 | 62.99 | 66.24 | 541.58 |
| CAMP | 54.91 | 64.74 | 67.50 | 547.71 |
| MFRGN | 12.22 | 20.75 | 41.14 | 1485.79 |
| Game4Loc | 49.57 | 59.68 | 65.53 | 612.22 |
| MMGeo | 44.58 | 55.35 | 61.25 | 681.87 |
| CVcities | 52.56 | 63.77 | 68.93 | 486.36 |
| InfoGeo | $56.88_{+1.97}$ | $67.00_{+2.26}$ | $70.60_{+1.67}$ | $420.08_{-66.28}$ |
| InfoGeo* | $57.90_{+2.99}$ | $67.88_{+3.14}$ | $71.18_{+2.25}$ | $416.75_{-69.61}$ |

*Table 3.* Comparisons on different weather cross-view transfer tasks (evaluated on multiple diverse weather scenarios).

| Model | Fog-Rain | | Rain-Snow | | Fog-Snow | |
|---|---|---|---|---|---|---|
| | R@1↑ | AP↑ | R@1↑ | AP↑ | R@1↑ | AP↑ |
| Sample4Geo | 37.39 | 44.13 | 45.80 | 51.58 | 26.68 | 33.05 |
| DAC | 73.23 | 76.72 | 68.97 | 73.07 | 45.02 | 50.57 |
| CVcities | 75.11 | 79.06 | 85.23 | 87.79 | 72.33 | 76.51 |
| InfoGeo | **81.55** | **84.66** | **90.30** | **91.92** | **76.81** | **81.25** |
| InfoGeo* | **82.80** | **85.49** | **92.29** | **93.56** | **79.95** | **82.95** |

view. As shown in Table 3, the reported results are averaged across different heights, covering three extreme scenarios: **Fog-Rain**, **Fog-Snow**, and **Rain-Snow**.

Our method consistently outperforms prior approaches across all weather conditions, achieving an average improvement of 6.0%. Notably, InfoGeo shows clear gains under severe transition **Fog-Snow**, surpassing the baseline CVcities by about 7.6% and 6.4% at R@1 and AP, respectively, demonstrating its ability to preserve task-relevant structural cues under appearance variations.

Moreover, we validate this observation through PCA visualizations (see Figure 4 (b)) of SUES-200@150 m, where object-centric representations disentangle discriminative instances such as buildings and roads from background clutter while filtering view-specific noise (e.g., snow). Cross-view concept affinities further indicate higher spatial-level consistency, confirming that InfoGeo effectively compresses weather-induced noise under challenging conditions.

*Table 4.* Computational cost evaluation of comparison methods.

| | Sample4Geo | DAC | CVcities | InfoGeo | InfoGeo* |
|---|---|---|---|---|---|
| Params (M) | 87.51 | 87.51 | 90.50 | 121.01 | 90.50 |
| FLOPs (G) | 72.60 | 80.22 | 91.67 | 105.63 | 91.70 |

across five UAV transfer scenarios: University-1652→SUE-200 / DenseUAV, DenseUAV→SUE-200 / University-1652, and GTA-V (cross-area). Compared with the baseline CVcities, the average R@1 improves from 90.27% to 95.07% on University-1652→SUE-200 and from 38.87% to 42.30% on University-1652→DenseUAV. While InfoGeo achieves 93.41% and 69.37% R@1 on DenseUAV→SUE-200 / University-1652, surpassing the baseline by 3.74% and 3.29%, respectively. On the challenging GTA-V dataset, InfoGeo outperforms the previous SOTA CAMP by 2.99% at R@1 and improves Dis@1 by over 30%. These consistent gains validate our design: object-centric representations capture view-invariant essentials, while RD integrates them into scene-level descriptors for more robust localization. As illustrated in Figure 4 (a), the qualitative results further show that our method effectively captures view-shared objects and exhibits high semantic consistency of the corresponding concepts across different viewpoints.

**Multi-Weather Robustness Evaluation.** We further conduct experiments on University-1652→SUES-200 under multi-weather settings (Zheng et al., 2024) in query (UAV)

**Complexity Analysis.** As shown in Table 4, our approach with RD significantly reduces computational costs, achieving similar complexity with the standard DINOv2-Base backbone, ensuring high inference efficiency while preserving the robust visual representations for localization.

## 5.4. Ablation Study

**Ablation of Main Components.** We perform an ablation study on individual components to verify our design, as shown in Table 5 (Comprehensive results on more altitudes are illustrated in Appendix Table 12).

*Table 5.* The ablations on different components of our proposed method evaluated on SUES@150 m. († indicates that the baseline model is trained using the same parameters as our method.)

| Model | University→SUES | | DenseUAV→SUES | |
|---|---|---|---|---|
| | **R@1↑** | **AP↑** | **R@1↑** | **AP↑** |
| Baseline | 82.73 | 85.60 | 80.20 | 83.24 |
| Baseline† | 86.10 | 88.28 | 83.28 | 86.14 |
| OCVA-only | 87.80 | 89.69 | 85.75 | 88.25 |
| + $\mathcal{L}_{cacs}$ | 88.82 | 90.61 | 85.97 | 88.40 |
| + $\mathcal{L}_{struct}$ | 90.87 | 92.41 | 88.20 | 89.33 |
| + RD | **91.80** | **93.18** | **88.70** | **90.73** |

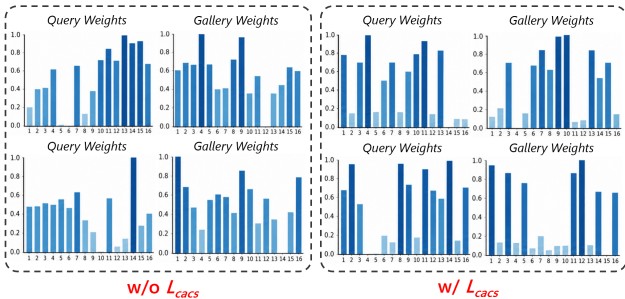

*Figure 5.* Visualizations on the impact of cross-view adaptive concept selection ($\mathcal{L}_{cacs}$) for the adaptive weights $\mathbf{W}_{cv}$ (trained on University-1652).

The introduction of OCVA module provides solid performance boost by incorporating object-centric semantics.

$\mathcal{L}_{cacs}$ further enhances results by filtering view-specific noise via cross-view correlations. We also conduct qualitative analysis to explore the impact of CACS on $\mathbf{W}_{cv}$ in Figure 5. The weight distributions indicate that $\mathcal{L}_{cacs}$ encourages more decisive and adaptive concept selection. Specifically, the learned weights become more polarized, suppressing many less relevant concepts toward zero while strongly activating a small subset of informative ones. Moreover, the activated concepts differ across query and gallery views, suggesting diverse cross-view concept selection.

In addition, adding $\mathcal{L}_{struct}$ (CSRR module) leads to more than 2% gain in R@1, validating the importance of object-

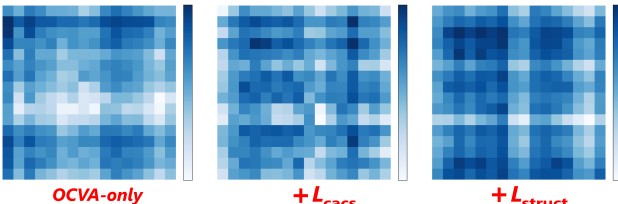

*Figure 6.* The cross-view spatial-level concept affinities of different components produced by decoding attention maps $\hat{\mathbf{A}}_d^{(v)}$.

centric concept structural alignment. These improvements are corroborated by the visualizations on spatial-level concept affinities in Figure 6 (University-1652→SUES-200@150 m), where $\mathcal{L}_{cacs}$ sharpens spatial attention by filtering view-specific noise and $\mathcal{L}$struct further enforces structural consistency.

Finally, incorporating RD yields peak performance (average gains of 0.7% and 1.5% in R@1 and AP) on both scenarios, demonstrating that relational distillation complements structural constraints.

Overall, the results confirm that each module contributes synergistically to the framework's generalization.

*Table 6.* The ablations on the loss design in the CSRR module.

| Method | University→SUES | | DenseUAV→SUES | |
|---|---|---|---|---|
| | **R@1↑** | **AP↑** | **R@1↑** | **AP↑** |
| w/o CSRR | 88.65 | 90.55 | 85.88 | 88.34 |
| *InfoNCE* | 88.20 | 90.03 | 86.64 | 88.77 |
| $\mathcal{L}_{struct}$ ($r=2$) | 89.22 | 91.00 | 86.85 | 89.06 |
| $\mathcal{L}_{struct}$ ($r=4$) | **91.80** | **93.18** | **88.70** | **90.73** |
| $\mathcal{L}_{struct}$ ($r=8$) | 86.42 | 88.51 | 85.75 | 88.11 |

**Effects of CSRR module.** Interestingly, CSRR module yields significant performance gains in ablation studies, motivating a comparative investigation of $\mathcal{L}_{struct}$ and the common *InfoNCE* loss as alternative objectives for cross-view concept alignment.

As shown in Table 6, we evaluate the effectiveness of the proposed structural loss $\mathcal{L}_{struct}$ across two benchmarks. The results confirm that the integration of CSRR with $\mathcal{L}_{struct}$ significantly outperforms the baseline, whereas the standard *InfoNCE* loss, relying on direct alignment of unordered features and ignoring concept-level structure, is less effective and may even degrade performance. This underscores the necessity of structural relation constraints for cross-view alignment. Besides, we investigate the sensitivity of the hyperparameter $r$. When $r$ is set to 4, it serves as the optimal bottleneck for semantic refinement, achieving superior performance. Increasing $r$ to 8 leads to a sharp performance decline, suggesting that overly coarse granularity collapses discriminative local features.

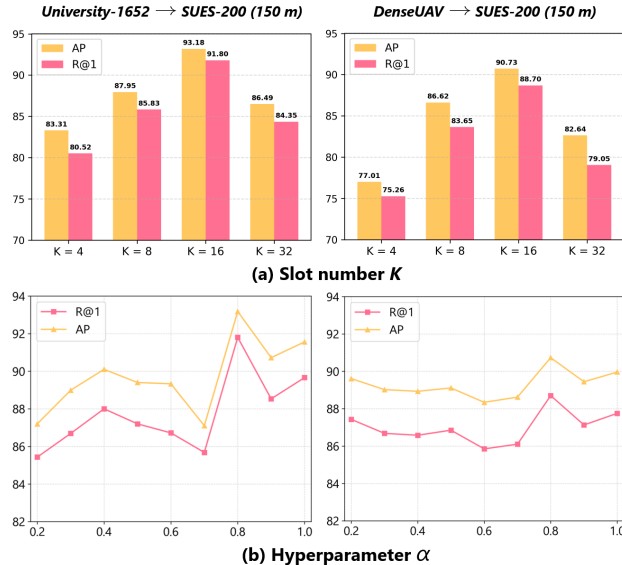

*Figure 7.* The sensitivity analysis of the hyperparameters $K$ and $\alpha$ in OCVA modules on two cross-dataset scenarios.

**Effects of OCVA module.** Finally, we conduct sensitivity analyses on both slot number $K$ and the weighting parameter $\alpha$ to evaluate the stability of OCVA module on SUES-200 (150 m) dataset, as illustrated in Figure 7 (a) and (b).

As $K$ varies from 4 to 16, it improves high-level semantic reconstruction into distinct object-centric representations, effectively avoiding slot collapse issue where excessively similar concepts. In contrast, the generalization performance is degraded when $K$ is 32, as excessive slots cause redundant discrete concepts, introducing noisy information that weakens discriminative cues. Thus, setting $K$ to 16 provides an optimal value in the module.

Meanwhile, the model achieves its best performance at $\alpha = 0.8$, with both metrics exhibiting consistent trends across scenarios. These results highlight the importance of balancing object-centric cues and global context, where an appropriate $\alpha$ emphasizes discriminative regions while preserving global spatial structure. In contrast, intermediate $\alpha$ values induce a suboptimal optimization trade-off, increasing uncertainty and exacerbating slot collapse, which ultimately degrades overall performance.

## 6. Conclusion

We introduce InfoGeo, a cross-view object-centric learning framework for CVGL, and provide an information-theoretic justification of its effectiveness. From the perspective of information bottleneck theory, we reformulate cross-view OCL with two complementary objectives: compressing view-specific noisy signals and maximizing view-invariant information. Specifically, we propose a cross-view adaptive concept selection module to suppress view-specific noise via cross-view collaborative knowledge, while concept structural relational reasoning captures inter-view structural relations to maximize the mutual information. Furthermore, the channel-wise FiLM blocks are employed for object-centric visual augmentation, allowing object-centric perceptual cues to be effectively incorporated into the global feature maps. Extensive experimental results demonstrate the robustness and effectiveness of InfoGeo for generalizable UAV-based cross-view geo-localization.

## Acknowledgement

This work was supported in part by Guangdong Science and Technology Department under Grant 2025A0505000062 and the "1+1+1 CUHK-CUHK(SZ)-GDSTC Joint Collaboration Fund - Grant 2025A0505000079".

## Impact Statement

This paper presents work whose goal is to advance the field of cross-view geo-localization. There are many potential societal consequences of our work, none which we feel must be specifically highlighted here.

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

**Appendix Outline.** This appendix presents the related works of cross-view geo-localization and multi-view object-centric learning, additional implementation details of the methodology, extended experiments, visualization, and further analysis of the proposed InfoGeo. The contents are organized as follows:

- § **Section A**: Related Work.

- § **Section B**: More Details on Methodology.

- § **Section C**: Extended Experimental Results.

- § **Section D**: Qualitative Results.

- § **Section E**: Discussion.

# A. Related Work

## A.1. Cross-View Geo-Localization

Recent image retrieval methods primarily focus on learning compact and discriminative global descriptors, while leveraging selective local feature aggregation to capture fine-grained visual cues. DELG (Cao et al., 2020), DOLG (Yang et al., 2021) jointly model global and local cues, while scalable frameworks such as CosPlace (Berton et al., 2022) and MixVPR (Ali-Bey et al., 2023) improve generalization via optimized global aggregation and large-scale training. Building upon these retrieval paradigms, cross-view geo-localization (CVGL) adapts image retrieval techniques to match ground-level images with aerial or satellite views under extreme viewpoint and geometric discrepancies.

Ground-to-aerial localization methods (Wu et al., 2026; Ouyang & Zhu, 2026) focus on mitigating the cross-view gap caused by extreme perspective changes. VIGOR (Zhu et al., 2021) reformulates CVGL beyond the idealized one-to-one correspondence assumption, explicitly modeling region-level ambiguity inherent in real-world localization scenarios. Subsequent works explore complementary directions to bridge the cross-view discrepancy, including view synthesis for representation regularization (Toker et al., 2021), geometry-aware alignment for pose-level localization (Shi et al., 2022), and simplified yet effective backbone designs tailored for cross-view matching (Zhu et al., 2023b). These approaches primarily target ground-level imagery and emphasize robustness to viewpoint and appearance variations.

More recent studies extend CVGL to UAV scenarios, where images are captured at varying altitudes and viewing angles, introducing additional scale and structural variability. Representative efforts emphasize patch-level correspondence modeling and self-distillation (Li et al., 2023), and integrate 3D structural representations into cross-view feature alignment (Zhang et al., 2025). However, UAV-based CVGL remains challenging due to altitude-dependent object density, increased background clutter, and limited availability of consistently discriminative objects, which often lead to degraded object-level correspondence and reduced generalization across various flight heights.

## A.2. Multi-View Object-Centric Learning

Learning object-centric scene representations is crucial for achieving structured understanding and abstraction of complex scenes. However, most existing methods focus only on single-view settings, resulting in single-view spatial ambiguities and leading to several failures or inaccuracies in representing 3D scene properties and in handling dynamic scenes with spatial structures (Li et al., 2020; 2021). DyMON (Li et al., 2021) extends multi-view object-centric learning to dynamic scenes via spatial-temporal factorization. While enhancing motion and temporal coherence modeling, representations may still entangle viewpoint variations and noise, as reconstruction-based disentanglement remains implicit. Ye et al. (Yuan et al., 2024) leverage multi-view consistency to maintain object permanence across unposed viewpoints. However, by relying on generative reconstruction for alignment, this approach lacks explicit control over separating view-invariant semantics from view-specific factors. Huang et al. (Huang et al., 2024b) innovatively propose actively selecting informative viewpoints to enhance efficiency and invariance. But this prioritizes view selection over determining which information to retain within object-centric representations. For cross-view visual reasoning, ObjectRelator (Fu et al., 2025) explicitly models object relations across ego-centric and exo-centric views, improving relational understanding under large viewpoint changes. However, it focuses on correspondence and relation reasoning rather than suppressing viewpoint-specific nuisances, which can limit robustness under severe appearance or structural variations.

Overall, existing methods rely on the methods of generative reconstruction, geometric consistency, or relational alignment to implicitly encourage shared structure across views, but lack an explicit mechanism to regulate information flow, leading to the entanglement of view-invariant object semantics with view-specific nuisances such as appearance changes, occlusions, or background clutter in dynamic or unconstrained environments.

## B. More Details on Methodology

### B.1. The Components of Object-Centric Learning

Most existing object-centric learning (OCL) models typically consist of three main components: a slot encoder, an aggregator, and a slot decoder. The detailed description of each component is provided as follows:

**Slot Encoder.** The encoder $\phi_e$ is responsible for extracting low-level and mid-level visual features from the input image. It is typically implemented using an MLP. The encoder maps the input feature maps $\mathbf{Z}$ into a set of $N$ feature vectors:

$$\mathbf{F} = \phi_e(\mathbf{Z}) = \{\mathbf{f}_i\}_{i=1}^N, \quad \mathbf{f}_i \in \mathbb{R}^D, \tag{21}$$

where $N = H \times W$ corresponds to the number of spatial locations and $D$ is the feature dimension. To preserve spatial information, positional encodings are commonly added to the extracted features:

$$\mathbf{f}_i \leftarrow \mathbf{f}_i + \mathrm{PE}(\mathbf{p}_i), \tag{22}$$

where $\mathbf{p}_i$ denotes the 2D spatial coordinates of the $i$-th feature and $\mathrm{PE}(\cdot)$ is a positional encoding function.

**Aggregator.** The aggregator $\phi_a$ aims to transform the unordered feature set $\mathbf{F}$ into a structured collection of object-centric representations, referred to as slots. Let $K$ denote the number of slots. The slots are initialized as a set of learnable or randomly sampled vectors:

$$\mathbf{S}^{(0)} = \{\mathbf{s}_k^{(0)}\}_{k=1}^K, \quad \mathbf{s}_k^{(0)} \sim \mathcal{N}(\mathbf{0}, \mathbf{I}). \tag{23}$$

Slot attention is applied iteratively to assign feature tokens to slots. At iteration $t$, the $i$-th attention weights $a_{ik}^{(t)}$ between features and slots are computed as:

$$a_{ik}^{(t)} = \frac{\exp\left((\mathbf{W}_q \mathbf{s}_k^{(t)})^\top (\mathbf{W}_k \mathbf{f}_i)\right)}{\sum_{k'=1}^K \exp\left((\mathbf{W}_q \mathbf{s}_{k'}^{(t)})^\top (\mathbf{W}_k \mathbf{f}_i)\right)}, \tag{24}$$

where $\mathbf{W}_q$ and $\mathbf{W}_k$ are learnable linear projections. The attention weights are normalized over the slot dimension, enforcing competition among slots for explaining each feature. Then, slot-wise aggregated updates are computed as

$$\mathbf{u}_k^{(t)} = \sum_{i=1}^N a_{ik}^{(t)} \mathbf{W}_v \mathbf{f}_i, \tag{25}$$

where $\mathbf{W}_v$ is a linear projection. Each slot is then updated using a recurrent update rule:

$$\mathbf{s}_k^{(t+1)} = \mathrm{Update}\big(\mathbf{s}_k^{(t)}, \mathbf{u}_k^{(t)}\big), \tag{26}$$

where $\mathrm{Update}(\cdot)$ is typically implemented using a gated recurrent unit followed by a feed-forward refinement. After $T$ iterations, the final slot representations are given by $\mathbf{S} = \{\mathbf{s}_k^{(T)}\}_{k=1}^K$.

**Slot Decoder.** The decoder $\phi_d$ maps each object-centric slot back to the pixel space to reconstruct the input image. For each slot $\mathbf{s}_k$, the decoder predicts a slot-level feature map $\hat{\mathbf{Z}}_k$ and a corresponding soft mask $\mathbf{M}_k$:

$$(\hat{\mathbf{Z}}_k, \mathbf{M}_k) = \mathrm{Dec}(\mathbf{s}_k), \quad k = 1, \ldots, K, \tag{27}$$

where $\hat{\mathbf{Z}}_k \in \mathbb{R}^{H \times W \times 3}$ and $\mathbf{M}_k \in [0,1]^{H \times W}$. The final reconstructed outputs are obtained via soft composition:

$$\hat{\mathbf{Z}} = \sum_{k=1}^K \mathbf{Z}_k \odot \hat{\mathbf{Z}}_k, \tag{28}$$

where $\odot$ denotes element-wise multiplication. The reconstruction $\hat{\mathbf{Z}}$ are compared against the input $\mathbf{Z}$ using a reconstruction loss, which provide the primary self-supervised learning signal for training all three components jointly.

*Table 7.* Mutual information analysis on training sets.

| Metric | University-1652 | DenseUAV | GTA-V |
|---|---|---|---|
| $I(Z_q, Z_g)$ | 0.3472 | 0.0797 | 0.1540 |
| $I(Z_q, Z_g \mid Y)$ | 0.0000 | 0.0000 | 0.0000 |
| $I(\hat{Z}_q, \hat{Z}_g \mid Y)$ | 0.0000 | 0.0000 | 0.0000 |

## B.2. Information-Bottleneck Theory for CVGL

In this section, we further provide more detailed information about the proof of the cross-view mutual information bound.

**The quantitative analysis on Cross-View Mutual Information.** In the context of CVGL, two views are acquired by different platforms with independent sensors and imaging conditions. Therefore, given the location identity $Y$, the remaining view-specific variations are assumed to be conditionally independent.

To empirically verify the conditional assumptions, we further conduct MI analyses on training sets in Table 7. $I(Z_q, Z_g)$ denotes the overall statistical dependence between query and gallery features, reflecting nontrivial global cross-view dependence. However, both the conditional mutual information $I(Z_q, Z_g \mid Y)$ using pretrained DINOv2 and $I(\hat{Z}_q, \hat{Z}_g \mid Y)$ using trained InfoGeo are near zero in all benchmarks, indicating that no statistically significant dependence remains across two views. These results validate that the conditional independence assumption is reasonable in practice and provide strong support for the Markov chain formulation:

$$Y \rightarrow \hat{Z}_g \rightarrow Z_q \rightarrow \hat{Z}_q. \tag{29}$$

The above results empirically support the assumptions of conditional independence, where $\hat{Z}_q \perp \hat{Z}_g \mid Y$.

**The Proof of Theorem 3.2.** Learning visual representations by mutual information maximization has emerged as an effective paradigm for preserving task-relevant semantics in deep networks (Hjelm et al., 2019), while prior work (Amjad & Geiger, 2019) has investigated the information bottleneck principle for deep neural network classification, characterizing representation learning as a Markov chain processing that progressively filters task-irrelevant information.

Based on the above analysis, we model the cross-view representation learning flow in an analogous Markov manner. As the geographic information $\mathbf{Y}$ is unobservable, and under the conditional independence assumption between $\hat{\mathbf{Z}}^q$ and $\hat{\mathbf{Z}}^g$, the representation learning process induces the Markov chain as:

$$\mathbf{Y} \rightarrow \hat{\mathbf{Z}}^g \rightarrow \mathbf{Z}^q \rightarrow \hat{\mathbf{Z}}^q. \tag{30}$$

By the Data Processing Inequality (DPI), which states that information cannot increase through a Markov chain, the above formulation naturally leads to the following information-theoretic characterization:

$$I(\hat{\mathbf{Z}}^{(v)}; \hat{\mathbf{Z}}^{(\bar{v})}) - I(\hat{\mathbf{Z}}^{(v)}; \mathbf{Y}) = I(\hat{\mathbf{Z}}^{(v)}; \hat{\mathbf{Z}}^{(\bar{v})} \mid \mathbf{Y}) \geq 0 \rightarrow I(\hat{\mathbf{Z}}^{(v)}; \hat{\mathbf{Z}}^{(\bar{v})}) \geq I(\hat{\mathbf{Z}}^{(v)}; \mathbf{Y}). \tag{31}$$

After that, we leverage this property to establish a rigorous upper bound on cross-view mutual information, which is presented in Theorem 3.2.

**Applying Theorem 3.2 to Cross-View OCL.** Analogously, this information-theoretic perspective can be extended to the optimization of cross-view object-centric learning, as illustrated in Equation (18). To formalize it, we consider a cross-view setting with two views $q$ and $g$. For each view $v \in \{q, g\}$, we assume the data generative process governed by the following Markov chain:

$$\hat{\mathbf{Z}}^{(\bar{v})} \rightarrow \mathbf{Z}^{(v)} \rightarrow \hat{\mathbf{Z}}^{(v)} \rightarrow \hat{\mathbf{A}}_d^{(v)} \rightarrow \mathbf{U}^{(v)}. \tag{32}$$

Based on this formulation, we show that maximizing the concept-level mutual information $I(\mathbf{U}^q; \mathbf{U}^g)$ serves as a principled surrogate for optimizing the semantic alignment objective $I(\hat{\mathbf{Z}}^q; \hat{\mathbf{Z}}^g)$. The details of the proof follow the steps below:

1. With the gallery-view Markov chain fixed, we analyze the dependence between $\mathbf{U}^q$ and the variables along the gallery-view chain. From the Markov chain $\hat{\mathbf{Z}}^q \rightarrow \hat{\mathbf{A}}_d^q \rightarrow \mathbf{U}^q$, we can define as:

$$I(\mathbf{U}^q; \mathbf{U}^g) \leq I(\hat{\mathbf{A}}_d^q; \mathbf{U}^g) \leq I(\hat{\mathbf{Z}}^q; \mathbf{U}^g) \tag{33}$$

2. Now, applying DPI to the second chain $\hat{\mathbf{Z}}^g \to \hat{\mathbf{A}}_d^g \to \mathbf{U}^g$ with respect to the variable $\hat{\mathbf{Z}}^q$, we observe:

$$I(\hat{\mathbf{Z}}^q; \mathbf{U}^g) \leq I(\hat{\mathbf{Z}}^q; \hat{\mathbf{A}}_d^g) \leq I(\hat{\mathbf{Z}}^q; \hat{\mathbf{Z}}^g) \tag{34}$$

3. By transitivity of the inequalities derived in steps 1 and 2, we obtain the following inequality:

$$I(\mathbf{U}^q; \mathbf{U}^g) \leq I(\hat{\mathbf{Z}}^q; \hat{\mathbf{Z}}^g) \tag{35}$$

This confirms that $I(\mathbf{U}^q; \mathbf{U}^g)$ serves as a lower bound on latent semantic correspondence, thereby providing a principled theoretical justification for optimizing concept-level mutual information in cross-view object-centric learning. Therefore, the first term in Equation (6) can be converted into the first term in Equation (18).

We further characterize the cross-view object-centric learning process through the following Markov chains:

$$\mathbf{Z}^q \to \mathbf{A}_d^q \to \hat{\mathbf{A}}_d^q, \quad \mathbf{Z}^g \to \mathbf{A}_d^g \to \hat{\mathbf{A}}_d^g. \tag{36}$$

By the data processing inequality, the conditional mutual information is upper-bounded as follows:

$$I(\hat{\mathbf{Z}}^q; \mathbf{Z}^q \mid \hat{\mathbf{Z}}^g) \leq I(\hat{\mathbf{A}}_d^q; \mathbf{A}_d^q \mid \hat{\mathbf{A}}_d^g). \tag{37}$$

The proposed framework does not explicitly minimize the second term in Equation (18). Besides, this term serves as a theoretical motivation under IB theory. In practice, we realize its effect by learning cross-view adaptive routing weights $\mathbf{W}_{\text{cv}}$ in CACS module, which selectively emphasize view-shared concepts while attenuating view-specific ones. Importantly, rather than explicitly optimizing $\mathbf{A}_d^q$, our framework introduces a Concept Router that generates cross-view adaptive weights $\mathbf{W}_{\text{cv}}$, which modulate the attention maps to obtain $\hat{\mathbf{A}}_d^q$. This reweighting mechanism implicitly constrains the information flow from $\mathbf{A}_d^q$ to $\hat{\mathbf{A}}_d^q$, thereby regulating the conditional mutual information at the concept level. This reweighting process induces the following Markov chain at the concept level:

$$\mathbf{Z}^{(v)} \to \mathbf{A}_d^{(v)} \to \mathbf{W}_{\text{cv}} \to \hat{\mathbf{A}}_d^{(v)}, \tag{38}$$

where $\mathbf{Z}^{(v)}$ denotes the latent semantic variables of view $v$. Since $\mathbf{W}_{\text{cv}}$ is solely determined by cross-view information, the transformation from $\mathbf{A}_d^{(v)}$ to $\hat{\mathbf{A}}_d^{(v)}$ acts as a view-aware bottleneck that selectively filters concept-level information.

Equation (9) encourages sparse yet diverse concept selection across views. By suppressing redundant or view-specific concepts while preserving view-consistent ones, the Concept Router effectively reduces $I(\hat{\mathbf{A}}_d^q; \mathbf{A}_d^q \mid \hat{\mathbf{A}}_d^g)$, thereby enforcing the compression of view-dependent nuisance information in accordance with the information bottleneck principle.

Overall, the objective $\mathcal{L}_{\text{info}}$ can be expressed in terms of entropy as follows:

$$\mathcal{L}_{\text{info}} = \underbrace{H(\mathbf{U}^{(\bar{v})} \mid \mathbf{U}^{(v)})}_{\mathcal{L}_{\text{struct}}} - \sum_{v \in \{g,q\}} \underbrace{H(\hat{\mathbf{A}}_d^{(v)} \mid \hat{\mathbf{A}}_d^{(\bar{v})}) - H(\hat{\mathbf{A}}_d^{(v)} \mid \hat{\mathbf{A}}_d^{(\bar{v})}, \mathbf{A}_d^{(v)})}_{\mathcal{L}_{\text{cacs}}}, \tag{39}$$

The first component is $\mathcal{L}_{\text{struct}}$, maximizes the mutual information $I(\mathbf{U}^g; \mathbf{U}^q)$ by minimizing the conditional entropy $H(\mathbf{U}^g \mid \mathbf{U}^q)$, thereby enforcing semantic consensus at the embedding level. To this end, we employ an orthogonal Procrustes loss, which constrains the embeddings to be aligned through an orthogonal transformation, preserving their intrinsic geometric structure.

The second component, corresponding to $\mathcal{L}_{\text{cacs}}$, serves as a collaborative regularizer that constrains the cross-view attention $\hat{\mathbf{A}}_d^{(v)}$. By minimizing the conditional mutual information $I(\hat{\mathbf{A}}_d^{(v)}; \mathbf{A}_d^{(v)} \mid \hat{\mathbf{A}}_d^{(\bar{v})})$, the framework encourages $\hat{\mathbf{A}}_d^{(v)}$ to retain only the information in $\mathbf{A}_d^{(v)}$ that is predictable from the cross-view counterpart, thereby suppressing view-specific redundancy. Expanding this objective in terms of entropy gives a tractable form suitable for optimization:

$$\mathcal{L}_{\text{cacs}} = I(\hat{\mathbf{A}}_d^{(v)}; \mathbf{A}_d^{(v)} \mid \hat{\mathbf{A}}_d^{(\bar{v})}) = H(\hat{\mathbf{A}}_d^{(v)} \mid \hat{\mathbf{A}}_d^{(\bar{v})}) - H(\hat{\mathbf{A}}_d^{(v)} \mid \hat{\mathbf{A}}_d^{(\bar{v})}, \mathbf{A}_d^{(v)}). \tag{40}$$

The formulation in Equation (40) is instantiated by $\mathcal{L}_{\text{cacs}}$ in Equation (9). We further provide a detailed analysis of the optimization process below:

Since $\hat{\mathbf{A}}_d^{(v)}$ is obtained by element-wise gating $\mathbf{A}_d^{(v)}$ with $\mathbf{W}_{\mathrm{cv}}$, the uncertainty of $\hat{\mathbf{A}}_d^{(v)}$ is fully governed by $\mathbf{W}_{\mathrm{cv}}$ when $\mathbf{A}_d^{(v)}$ is given. As a result, optimizing the conditional mutual information objective can be equivalently relaxed to regularizing the distribution of $\mathbf{W}_{\mathrm{cv}}$. Specifically, $\mathbb{E}[\mathbf{W}_{\mathrm{cv}} \odot (1 - \mathbf{W}_{\mathrm{cv}})]$ corresponds to the first term in Equation (40). This term constitutes a quadratic regularizer over the gating variables, which upper-bounds the conditional entropy of $\hat{\mathbf{A}}_d^{(v)}$ given $\hat{\mathbf{A}}_d^{(\bar{v})}$ under a Bernoulli parameterization. Minimizing this expectation suppresses high-variance and ambiguous gating responses, thereby driving $\mathbf{W}_{\mathrm{cv}}$ toward low-entropy, near-deterministic solutions that are consistent across views.

Meanwhile, maximizing $\mathrm{Var}(\mathbf{W}_{\mathrm{cv}})$ (the second term in Equation (40)) prevents degenerate solutions by promoting discriminative diversity across spatial locations, thereby retaining informative and view-shared activations. Together, these two terms constitute a tractable relaxation of the original information-theoretic objective, balancing uncertainty reduction and information preservation in a unified manner.

**Variational Mutual Information Estimation.** Directly maximizing mutual information across different views is generally intractable due to the unknown joint distribution. Following the variational perspective (Poole et al., 2019), the mutual information of the feature maps across two views is optimized via a tractable lower bound:

$$I(\hat{\mathbf{Z}}^q; \hat{\mathbf{Z}}^g) \geq \mathbb{E}_{p(\hat{\mathbf{z}}^q, \hat{\mathbf{z}}^g)} \left[ \log \frac{f_\theta(\hat{\mathbf{Z}}^q, \hat{\mathbf{Z}}^g)}{\mathbb{E}_{p(\hat{\mathbf{z}}^g)}[f_\theta(\hat{\mathbf{Z}}^q, \hat{\mathbf{Z}}^g)]} \right], \tag{41}$$

where $f_\theta(\cdot, \cdot)$ is a learnable function. In practice, this formulation subsumes commonly used contrastive objectives (e.g., InfoNCE (Van den Oord et al., 2018)), providing a low-variance and scalable estimator of mutual information.

## B.3. Concept Graph Relation Construction

We further detail the construction of concept graphs $\mathbf{G}^{(v)}$. $\tilde{\mathbf{G}}^{(v)}$ are obtained by computing the cosine similarities between the decoding attention maps $\hat{\mathbf{A}}_d^{(v)}$ as:

$$\tilde{G}_{ij}^{(v)} = \frac{\langle \hat{\mathbf{A}}_{d,i}^{(v)}, \hat{\mathbf{A}}_{d,j}^{(v)} \rangle}{\|\hat{\mathbf{A}}_{d,i}^{(v)}\|_2 \|\hat{\mathbf{A}}_{d,j}^{(v)}\|_2}, \tag{42}$$

where $\hat{\mathbf{A}}_{d,i}^{(v)} \in \mathbb{R}^{H \times W}$ denotes the decoding attention map corresponding to the $i$-th concept in view $v$. The resulting matrix $\tilde{\mathbf{G}}^{(v)} \in \mathbb{R}^{K \times K}$ represents the pairwise semantic similarity among concepts within the same view.

However, the values of cosine similarity lie in $[-1, 1]$, we apply an affine transformation to ensure non-negativity:

$$\mathbf{G}_{ij}^{(v)} = \frac{\tilde{\mathbf{G}}_{ij}^{(v)} + 1}{2}, \tag{43}$$

followed by clipping to eliminate numerical artifacts below zero. This guarantees that $\mathbf{G}^{(v)}$ is symmetric and element-wise non-negative, satisfying the requirements of graph Laplacian construction.

From an information-theoretic perspective, this construction can be interpreted as an implicit information bottleneck imposed on the slot representations. Specifically, projecting the slot graph onto the subspace spanned by the lowest-frequency eigenvectors yields a representation $\mathbf{U}^{(v)}$ that retains information most relevant to the underlying semantic identity $\mathbf{Y}$, while discarding high-complexity variations that do not persist across views. Formally, $\mathbf{U}^{(v)}$ can be viewed as a low-dimensional sufficient statistic satisfying:

$$I(\mathbf{U}^{(v)}; \mathbf{Y}) \approx I(\mathbf{S}^{(v)}; \mathbf{Y}), \quad I(\mathbf{S}^{(v)}; \mathbf{Z}^{(v)} \mid \mathbf{U}^{(v)}) \approx 0, \tag{44}$$

where $\mathbf{Z}^{(v)}$ denotes the input latent factors of view $v$. This aligns with the principle of minimal sufficient representation, in which the spectral embedding preserves view-shared relational semantics while filtering out view-dependent nuisance factors. Consequently, this structured alignment acts as a cross-view information bottleneck, enforcing that only concept-level relational structures consistent across views are preserved.

## B.4. Relational Distillation

While OCVA incorporates object-centric perception into visual representations, directly employing slot-attention modules during inference would introduce additional computational complexity. To tackle this issue, we propose *Relational Distillation* (RD), which transfers object-centric relational knowledge into a vanilla backbone during training.

Specifically, given the original feature maps $\mathbf{Z}^q$ and $\mathbf{Z}^g$ from the query and gallery views, as well as their object-centric augmented counterparts $\tilde{\mathbf{Z}}^q$ and $\tilde{\mathbf{Z}}^g$, we obtain their corresponding scene-level descriptors $\mathbf{f}^{(v)}$ and $\tilde{\mathbf{f}}^{(v)}$ via a view-shared feature aggregation layer.

Rather than enforcing point-wise feature alignment, RD distills the structural knowledge among samples by matching the pairwise relationships. Concretely, given $N$ samples in view $v$, we construct a pairwise relational matrix $\mathbf{R}^{(v)}(\cdot) \in \mathbb{R}^{N \times N}$ in view $v$ through Euclidean distances:

$$\mathbf{R}_{ij}^{(v)}(\mathbf{f}) = \|\mathbf{f}_i^{(v)} - \mathbf{f}_j^{(v)}\|_2. \tag{45}$$

The module encourages the relational geometry of the original features to match that in the augmented features:

$$\mathcal{L}_{\text{distill}} = \frac{1}{2} \sum_{v \in \{q,g\}} \|\mathbf{R}(\mathbf{f}^{(v)}) - \mathbf{R}(\tilde{\mathbf{f}}^{(v)})\|_2. \tag{46}$$

Consequently, RD retains object-centric representation while ensuring computational efficiency by avoiding the usage of OCL modules during inference.

### B.5. Channel-wise FiLM Block

To effectively integrate object-centric semantics into the global representations, we employ a modified version of the Feature-wise Linear Modulation (FiLM) (Perez et al., 2018). Unlike the standard affine formulation, our implementation utilizes a *scaling-only* mechanism. This design choice focuses on dynamic feature selection, allowing the discovered concepts to amplify or suppress specific channels in the global feature map without introducing additive bias that might shift the semantic grounding.

Let $\mathbf{v}_h^{(v)} \in \mathbb{R}^{C_{\text{cond}}}$ denote the affinity vector extracted from view $v$, which encodes object-centric semantic information distilled by the proposed cross-view OCL framework. This vector is used as a conditioning signal to adaptively modulate the global feature representation via a channel-wise Feature-wise Linear Modulation (FiLM) mechanism.

**Generation of Channel-wise Scaling Coefficients.** To inject object-centric priors into the global representation, we first project the conditioning vector $\mathbf{v}_h^{(v)}$ into the channel space of the global feature map. Specifically, a learnable mapping $f_\theta : \mathbb{R}^{C_{\text{cond}}} \to \mathbb{R}^C$ is employed to generate channel-wise modulation coefficients:

$$\gamma = \sigma\left(f_\theta(\mathbf{v}_h^{(v)})\right), \tag{47}$$

where $f_\theta$ is implemented as a linear projection or a shallow multi-layer perceptron (MLP). The activation function $\sigma(\cdot)$ (e.g., Sigmoid) constrains the scaling coefficients $\gamma \in \mathbb{R}^C$ to a bounded range, ensuring stable modulation. Each element $\gamma_i$ can be interpreted as an adaptive importance weight that reflects the relevance of the $i$-th channel with respect to the object-centric concepts present in view $v$.

**Channel-wise Multiplicative Modulation.** Given the global feature map $\mathbf{Z}^{(v)} \in \mathbb{R}^{C \times H \times W}$ extracted from the backbone network, we apply channel-wise multiplicative modulation using the generated scaling coefficients. For each channel $i \in \{1, \ldots, C\}$, the modulated feature map $\hat{\mathbf{Z}}_i^{(v)}$ is computed as:

$$\hat{\mathbf{Z}}_i^{(v)} = \mathbf{Z}_i^{(v)} \odot \left(1 + \gamma_i^{(v)}\right). \tag{48}$$

Equivalently, the modulation process can be expressed in a compact vectorized form as:

$$\text{FiLM}_{\text{scale}}\left(\mathbf{Z}^{(v)}, \mathbf{v}_h^{(v)}\right) = \gamma(\mathbf{v}_h^{(v)}) \odot \mathbf{Z}^{(v)}, \tag{49}$$

where $\odot$ denotes element-wise multiplication with broadcasting along the spatial dimensions.

This formulation enables the model to selectively amplify channels that are consistent with view-shared object-centric semantics, while suppressing channels dominated by view-specific or noisy factors. As a result, the modulated representation is encouraged to emphasize view-invariant structural cues, thereby improving robustness and generalization in CVGL.

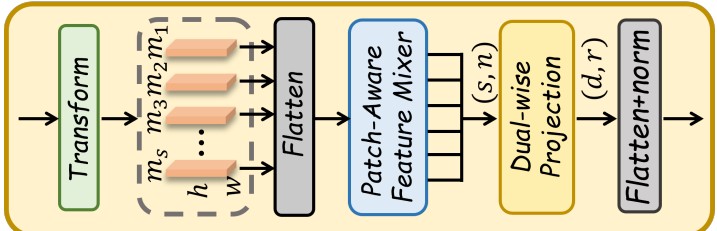

*Figure 8.* The detailed structures of the feature aggregation layer.

## B.6. Feature Aggregation Layer

The detailed structures of the feature aggregation layer is illustrated in Figure 8. This layer adopts a MLP-based aggregation strategy to iteratively inject global context into feature maps extracted from the pre-trained DINOv2 backbone. Specifically, the aggregation process is implemented through a stack of isotropic MLP blocks. In the Patch-Aware Feature Mixer, each block comprises LayerNorm, a linear projection, and an activation function, and we employ four such blocks sequentially. The Dual-wise Projection consists of two separate components: depth-wise and row-wise projections, each implemented with a single MLP layer. Specifically, the depth-wise projection outputs feature vectors of 512 dimensions, while the row-wise projection produces 4 rows, facilitating comprehensive spatial-context encoding.

## B.7. Optimization and Inference

**Optimization for cross-view feature alignment.**    In the CVGL task, the InfoNCE loss serves as the primary supervision signal for learning discriminative cross-view representations. It is designed to maximize the similarity between matched query-gallery pairs while simultaneously pushing apart non-matching pairs, effectively approximating the mutual information between views in a tractable manner (Van den Oord et al., 2018). Formally, the loss is defined as:

$$\mathcal{L}_{\text{align}}(\tilde{\mathbf{f}}^q, \tilde{\mathbf{f}}^g) = -\log \frac{\exp\left(\tilde{\mathbf{f}}^q \cdot \tilde{\mathbf{f}}^g_+ / \tau\right)}{\sum_{i=1}^{R} \exp\left(\tilde{\mathbf{f}}^q \cdot \tilde{\mathbf{f}}^g_i / \tau\right)}, \tag{50}$$

where $\tilde{\mathbf{f}}^q$ represents the feature embedding of the query image, and $\tilde{\mathbf{f}}^g$ denotes the set of gallery embeddings containing one positive sample $\tilde{\mathbf{f}}^g_+$ corresponding to the same location and $R-1$ negative samples. The temperature parameter $\tau$ modulates the concentration of the similarity distribution, controlling the extent to which the model emphasizes hard negatives. By employing this contrastive objective, the network is encouraged to learn view-invariant representations that are robust to perspective, illumination, and occlusion variations, which is crucial for achieving reliable cross-view geo-localization.

**Inference Process.**    For different networks used in our method, we adopt structure-specific inference strategies for evaluation. In particular, the Visual Extractor encapsulates all modules involved in the inference process. As illustrated in Figure 9, Relational Distillation simplifies the inference pipeline by transferring the relational knowledge from OCL and OCVA into the vanilla descriptors, thereby avoiding the additional computational complexity used in these modules.

## C. More Details and Results

### C.1. More Details on Experimental Settings

In the OCL components, the slot encoder maps the input visual features from 768 to 1024 dimensions. For unsupervised object discovery in the aggregator, we adopt the BO-QSA optimization strategy (Jia et al., 2023). The slot decoder is implemented using a Transformer architecture with four attention heads and four stacked layers (Vaswani et al., 2017), where the feed-forward network has a hidden dimension of 1536. For the iterative refinement process, $T$ is set to 3 for unconditional queries (Locatello et al., 2020) while $T = 1$ for conditional queries (Kipf et al., 2022).

For the proposed CACS module, the cross-view attention block consists of four attention heads to model fine-grained cross-view interactions, while the concept router is realized as a single-layer MLP for adaptive concept selection. For the channel-wise FiLM modulation, the conditioning function $\mathcal{G}(\cdot)$ is implemented as a single-layer MLP.

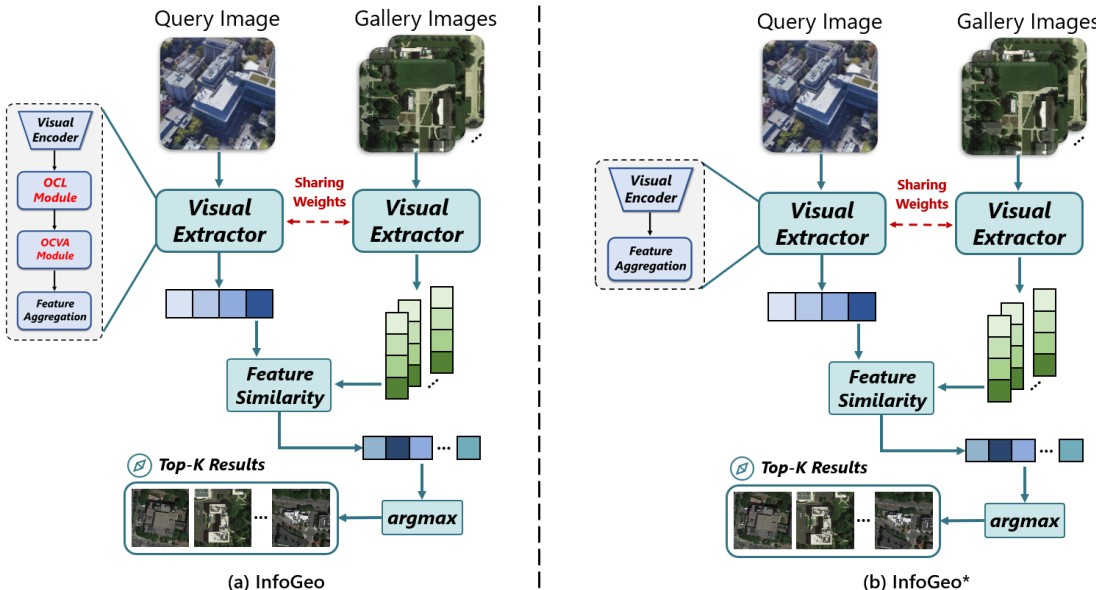

*Figure 9.* Comparison on the network structures of our proposed work in the inference stage. (a) InfoGeo w/o Relational Distillation, (b) InfoGeo w/ Relational Distillation.

*Table 8.* Statistical analysis of the UAV training datasets.

| Datasets | Altitude Range | IDs | Train Drone / Satellite | Test Drone / Satellite |
|---|---|---|---|---|
| University-1652 | 121m∼256m | 701 | 37,854 / 701 | 51,355 / 951 |
| SUES-200 | 150m | 120 | 6,000 / 120 | 4,000 / 120 |
| | 200m | 120 | 6,000 / 120 | 4,000 / 120 |
| | 250m | 120 | 6,000 / 120 | 4,000 / 120 |
| | 300m | 120 | 6,000 / 120 | 4,000 / 120 |
| DenseUAV | 80m∼100m | 2,256 | 6,768 / 13,536 | 2,331 / 18,193 |
| GTA-V (cross-area) | 80m∼650m | - | 15,693 / 14,640 | 18,070 / 14,640 |

The MixVPR-based aggregation layer is configured with a mix depth of 2 and an output row number of 4. During retrieval, the final scene-level descriptor is set to a dimensionality of 4096 to balance representational capacity and retrieval efficiency.

For optimization, we employ dataset-specific warm-up strategies to stabilize training across different benchmarks. Specifically, the warm-up ratios for University-1652, DenseUAV, and GTA-V are set to 0.1, 0.15, and 0.25, respectively.

### C.2. More Details on UAV benchmarks

We evaluate our method on four UAV-based benchmarks with varying scene complexity and viewpoint diversity. The detailed statistics on the number of training and testing images, altitude ranges, and identity counts for each dataset in Table 8. **University-1652** contains drone-satellite image pairs of university buildings captured at altitudes from 121 m to 256 m. **SUES-200** provides multi-altitude drone captures at 150 m, 200 m, 250 m, and 300 m, increasing viewpoint diversity and evaluation difficulty. **DenseUAV** targets drone self-localization under dense spatial layouts and fine-grained location ambiguity, with flights ranging from 80 m to 100 m. **GTA-V** is a more challenging benchmark covering altitudes from 80 m to 650 m, diverse scene types, and viewpoints, supporting both same-area and cross-area evaluation.

### C.3. Additional Experimental Results on Domain Generalization

In this section, we present additional experimental results on SUES-200 to further evaluate the domain generalization capability of our method under diverse cross-domain settings, as shown in Table 9 and Table 10. Across different flight altitudes in SUES-200, the results consistently demonstrate that our method maintains stable and effective performance,

*Table 9.* Performance (%) Comparison on University-1652→SUES-200 under different heights.

| Methods | Venue | University→SUES150 | | University→SUES200 | | University→SUES250 | | University→SUES300 | |
|---|---|---|---|---|---|---|---|---|---|
| | | R@1 | AP | R@1 | AP | R@1 | AP | R@1 | AP |
| MCCG | TCVST'23 | 57.62 | 62.80 | 66.83 | 71.60 | 74.25 | 78.35 | 82.55 | 85.57 |
| Sample4Geo | ICCV'23 | 70.05 | 74.93 | 80.68 | 83.90 | 87.35 | 89.72 | 90.03 | 91.91 |
| DAC | TCVST'24 | 76.65 | 80.56 | 86.45 | 89.00 | 92.95 | 94.18 | 94.53 | 95.45 |
| CAMP | TGRS'24 | 78.90 | 82.38 | 86.83 | 89.28 | 91.95 | 93.63 | 95.68 | 96.65 |
| MFRGN | ACM MM'24 | 65.90 | 70.93 | 74.28 | 78.38 | 85.05 | 87.61 | 87.88 | 90.06 |
| EM-CVGL | TGRS'24 | 60.03 | 65.69 | 71.50 | 76.17 | 80.35 | 83.85 | 85.93 | 88.34 |
| Game4Loc | AAAI'25 | 79.03 | 82.50 | 86.03 | 88.37 | 90.25 | 92.08 | 91.70 | 93.32 |
| MMGeo | ICCV'25 | 73.08 | 77.45 | 82.73 | 85.92 | 90.33 | 92.03 | 93.90 | 95.01 |
| CVcities | JSTARS'24 | 82.73 | 85.60 | 90.08 | 91.63 | 94.35 | 95.12 | 93.90 | 94.74 |
| InfoGeo | Ours | **90.87** | **92.53** | **94.20** | **95.12** | **96.08** | **96.30** | **96.31** | **96.75** |
| InfoGeo* | Ours | **91.80** | **93.18** | **95.40** | **96.15** | **96.58** | **97.08** | **96.48** | **97.00** |

*Table 10.* Performance (%) Comparison on DenseUAV→SUES-200 under different heights.

| Methods | Venue | DenseUAV→SUES150 | | DenseUAV→SUES200 | | DenseUAV→SUES250 | | DenseUAV→SUES300 | |
|---|---|---|---|---|---|---|---|---|---|
| | | R@1 | AP | R@1 | AP | R@1 | AP | R@1 | AP |
| MCCG | TCVST'23 | 64.72 | 69.64 | 78.95 | 82.27 | 87.38 | 89.54 | 89.30 | 91.27 |
| Sample4Geo | ICCV'23 | 67.00 | 71.54 | 84.18 | 86.62 | 84.93 | 87.41 | 79.65 | 81.86 |
| DAC | TCVST'24 | 82.43 | 85.19 | 87.23 | 89.35 | 89.70 | 91.52 | 86.38 | 88.72 |
| CAMP | TGRS'24 | 72.30 | 76.12 | 89.63 | 91.59 | 90.53 | 92.35 | 94.68 | 95.66 |
| MFRGN | ACM MM'24 | 61.50 | 66.09 | 62.30 | 66.97 | 72.33 | 75.58 | 72.73 | 75.90 |
| EM-CVGL | TGRS'24 | 40.45 | 46.23 | 51.78 | 57.44 | 62.67 | 67.64 | 70.82 | 74.83 |
| Game4Loc | AAAI'25 | 53.20 | 58.94 | 66.53 | 70.85 | 73.85 | 77.25 | 75.18 | 78.55 |
| MMGeo | ICCV'25 | 76.15 | 79.48 | 77.03 | 80.23 | 80.02 | 82.90 | 82.98 | 85.59 |
| CVcities | JSTARS'24 | 80.20 | 83.24 | 89.73 | 91.54 | 93.83 | 94.87 | 94.90 | 95.65 |
| InfoGeo | Ours | **88.18** | **90.33** | **93.10** | **94.21** | **96.00** | **96.60** | **96.25** | **96.70** |
| InfoGeo* | Ours | **88.70** | **90.73** | **93.20** | **94.33** | **95.55** | **96.23** | **96.18** | **96.77** |

indicating its robustness to altitude-induced domain shifts. Nevertheless, when evaluated at a larger spatial separation of 300 m in the University→SUES-200 setting, we observe a relative performance drop compared to 250 m. This phenomenon can be attributed to an intrinsic limitation of object-centric learning in handling concept scalability. As the flight altitude increases, the number of observable objects grows substantially, while the set of objects with strong discriminative power remains limited. The resulting proliferation of weakly informative or noisy object signals dilutes the contribution of salient concepts, leading to degraded performance.

## C.4. Additional Experimental Results on Multi-Weather Transfer Task

As illustrated in Table 11, we further report additional experimental results under multi-weather settings to comprehensively evaluate the robustness and generalization capability of our method when confronted with diverse and challenging weather-induced appearance variations.

In such composite extreme weather conditions, particularly at lower flight altitudes (lower than 250 m), InfoGeo consistently outperforms the CVcities baseline, achieving an R@1 improvement of up to 15%. This substantial gain indicates that our information-theortic object-centric learning method effectively mitigates severe appearance distortions caused by compounded weather perturbations, thereby preserving discriminative and spatially consistent cues that are critical for accurate geo-localization under challenging low-altitude settings (150 m).

## C.5. More Ablation Studies

### C.5.1. ABLATIONS ON DIFFERENT COMPONENTS WITH MORE SCENARIOS

To validate the effectiveness of InfoGeo, we further evaluate its performance across different scenarios by comparing results on SUES-200 and SUES-300, enabling an analysis of model behavior under varying altitude conditions. As illustrated

*Table 11.* Performance (%) Comparison under different weather transfer settings on University-1652→SUES-200.

| Methods | University→SUES150 | | University→SUES200 | | University→SUES250 | | University→SUES300 | |
|---|---|---|---|---|---|---|---|---|
| | R@1 | AP | R@1 | AP | R@1 | AP | R@1 | AP |
| **(a) Fog-Rain** | | | | | | | | |
| Sample4Geo | 26.63 | 33.78 | 35.95 | 42.65 | 41.67 | 48.23 | 45.33 | 51.87 |
| DAC | 61.48 | 66.38 | 72.15 | 75.85 | 78.68 | 81.46 | 80.62 | 83.20 |
| CVcities | 65.10 | 70.34 | 75.17 | 79.08 | 80.33 | 83.52 | 79.85 | 83.30 |
| InfoGeo | **78.90** | **82.33** | **80.62** | **83.67** | **83.12** | **86.05** | **83.55** | **86.58** |
| InfoGeo* | **80.23** | **83.45** | **83.12** | **85.78** | **83.85** | **86.45** | **83.98** | **86.26** |
| **(b) Fog-Snow** | | | | | | | | |
| Sample4Geo | 21.40 | 27.66 | 25.65 | 31.99 | 29.25 | 35.64 | 30.42 | 36.92 |
| DAC | 39.40 | 45.17 | 45.25 | 50.60 | 46.53 | 52.21 | 48.92 | 54.31 |
| CVcities | 63.43 | 68.71 | 72.15 | 76.05 | 77.30 | 81.16 | 76.42 | 80.11 |
| InfoGeo | **74.60** | **78.92** | **76.63** | **80.16** | **77.50** | **81.52** | **76.50** | **80.00** |
| InfoGeo* | **78.65** | **81.89** | **80.48** | **83.46** | **81.35** | **84.09** | **79.90** | **82.82** |
| **(c) Rain-Snow** | | | | | | | | |
| Sample4Geo | 35.55 | 41.77 | 44.50 | 50.38 | 50.27 | 55.95 | 52.90 | 58.22 |
| DAC | 59.78 | 64.62 | 67.40 | 71.57 | 74.05 | 77.72 | 74.65 | 78.37 |
| CVcities | 75.60 | 79.54 | 85.00 | 87.53 | 89.77 | 91.68 | 90.57 | 92.42 |
| InfoGeo | **87.70** | **89.92** | **89.93** | **91.66** | **91.23** | **92.60** | **92.35** | **93.51** |
| InfoGeo* | **89.55** | **91.32** | **92.58** | **93.84** | **93.53** | **94.58** | **93.48** | **94.49** |

*Table 12.* Ablation study on SUES at 200 m and 300 m. (†: baseline trained with the same settings as our method.)

| Model | University→SUES (200m) | | DenseUAV→SUES (200m) | | University→SUES (300m) | | DenseUAV→SUES (300m) | |
|---|---|---|---|---|---|---|---|---|
| | R@1↑ | AP↑ | R@1↑ | AP↑ | R@1↑ | AP↑ | R@1↑ | AP↑ |
| Baseline | 90.08 | 91.63 | 89.73 | 91.54 | 93.90 | 94.74 | 94.90 | 95.65 |
| Baseline† | 91.23 | 92.70 | 90.65 | 92.35 | 94.80 | 95.56 | 95.53 | 96.02 |
| OCVA-only | 92.96 | 94.14 | 91.22 | 92.89 | 95.60 | 96.33 | 96.02 | 96.41 |
| + $\mathcal{L}_{cacs}$ | 93.72 | 94.78 | 91.35 | 93.05 | 95.83 | 96.51 | 96.11 | 96.56 |
| + $\mathcal{L}_{struct}$ | 94.05 | 95.10 | 92.93 | 94.16 | 96.31 | 96.75 | **96.25** | 96.70 |
| + RD | **95.40** | **96.15** | **93.20** | **94.33** | **96.48** | **97.00** | 96.18 | **96.77** |

in Table 12, it shows that each component consistently contributes to the overall performance across different heights in SUES-200. Specifically, in University→SUES (200 m), R@1 improves from 91.23% to 95.40% (+4.17%). Consistently, for DenseUAV→SUES (300 m), the full model achieves a peak AP of 96.77%. These quantitative results demonstrate the robustness of InfoGeo and superior cross-view localization performance across varying altitudes and datasets.

However, when relation distillation (RD) is introduced, we observe a noticeable drop in R@1 performance at 300 m. At this scale, the number of observable objects increases substantially while only a limited subset remains discriminative. Consequently, object relations become noisy and unstable, and distilling such relations enforces spurious structural constraints, which conflict with the object-centric compression mechanism and lead to degraded retrieval accuracy.

C.5.2. SENSITIVITY ANALYSIS ON HYPER-PARAMETER $\lambda_1$, $\lambda_2$ AND $\lambda_3$

To explore the influence of the different components in our overall objective, we conduct a sensitivity analysis on the three hyperparameters: $\lambda_1$, $\lambda_2$, and $\lambda_3$, as illustrated in Figure 10. $\lambda_1$ controls the reconstruction loss in the OCL module, $\lambda_2$ weights the Information-Theoretic based Cross-View Object-Centric Learning branch, and $\lambda_3$ regulates the contribution of the Relational Distillation module.

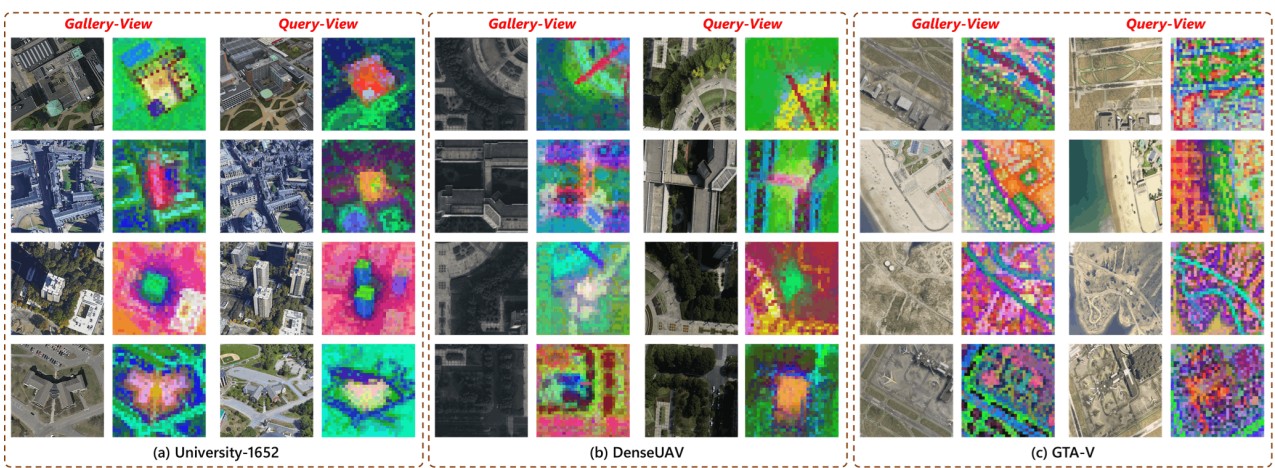

*Figure 10.* The ablations on the three hyperparameters for optimization across different scenarios.

We note that $\lambda_1$ and $\lambda_2$, serving as auxiliary objectives, must be carefully constrained to moderate values. Excessive weighting of either term leads to notable performance degradation, as optimization becomes dominated by auxiliary constraints rather than retrieval-focused supervision. In particular, an overly large $\lambda_1$ excessively enforces the reconstruction objective, causing multiple slots to converge to similar representations, which reduces object diversity and impairs discriminative capacity. By contrast, $\lambda_3$ requires a relatively higher coefficient to effectively guide relational distillation. Adequate weighting of $\lambda_3$ promotes consistent inter-object relations across views, enhancing representation robustness without inducing the adverse effects associated with overly strong auxiliary terms.

## D. Qualitative Results

### D.1. Visualization on Object-Centric Representations

We further present additional PCA visualizations of object-centric representations on the three benchmarks in Figure 11, where the features are extracted using the model weights trained on the corresponding training sets of each benchmark, ensuring that the visualizations faithfully reflect the learned representation characteristics under each dataset setting.

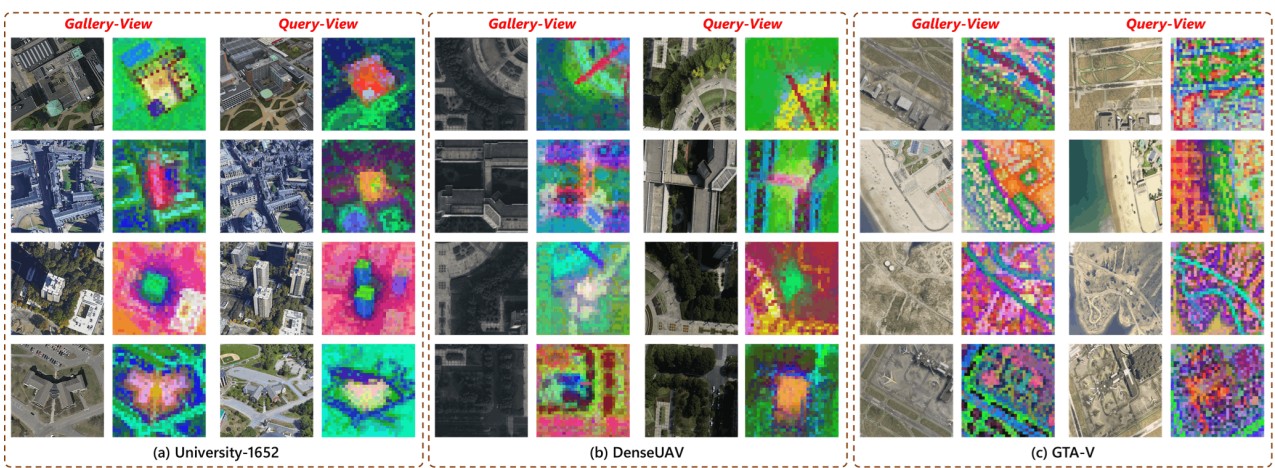

*Figure 11.* The PCA visualization of feature maps between different UAV benchmarks.

For University-1652 and DenseUAV, which mainly consist of urban scenes, geo-localization is largely determined by fine-grained structural cues such as building rooftops and road network layouts. By explicitly modeling object-centric representations, InfoGeo can consistently align these view-shared and view-invariant semantic structures across viewpoints, thereby reducing cross-view discrepancies and enhancing representation coherence. In contrast to the largely one-to-one cross-view correspondences ("perfect matching") observed in University-1652 and DenseUAV, GTA-V does not exhibit such superior results, making the localization task substantially more challenging. In this setting, discriminative cues primarily arise from higher-level structural information, such as road topology and building boundary geometries, rather than precise object-level alignments.

### D.2. Visualization on Multi-Weather Settings

As shown in Figure 12, we provide a qualitative comparison of feature map visualizations to analyze the robustness of different methods under multi-weather conditions.

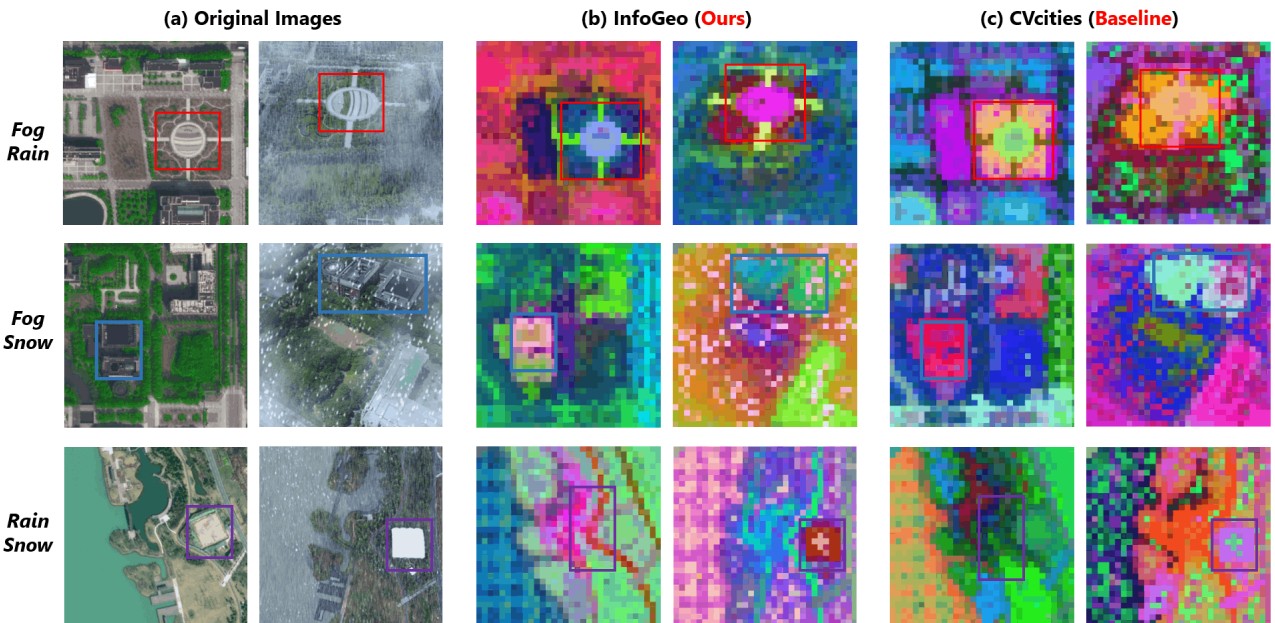

*Figure 12.* Comparison of the visualization on feature maps between InfoGeo and Baseline under Multi-Weather settings. The bounding boxes denote the view-shared objects across different viewpoints.

Under extreme weather conditions, baseline models fail to maintain view-consistent representations. For instance, in the fog-rain scenario, the discriminative roundabout-crossroad structure is poorly localized by the baseline and suffers from feature ambiguity. In contrast, InfoGeo effectively distinguishes the geometric semantics of roundabout-cross intersections, leveraging them as key discriminative cues. Similarly, in the rain-snow setting, our method extracts more robust information for localization, where road network structures and building rooftops are consistently highlighted in UAV views. In the more challenging fog-snow scenario, global representations learned by baseline models fail to effectively distinguish two distant buildings and lack discriminative capability for the green building in the foreground. In contrast, the proposed object-centric representations successfully separate relevant building instances and preserve effective key visual clues for reliable geo-localization.

### D.3. Failure Case Retrieval Results

To investigate the underlying failure patterns and model limitations, we present retrieval results (see Figure 13) for models trained on University-1652 and evaluated on SUES-200 at two UAV altitudes: 150 m and 300 m. InfoGeo demonstrates substantial performance gains at 150 m, whereas improvements over the baseline are less pronounced at 300 m, motivating a detailed analysis of retrieval behaviors across these altitude settings.

Figure 13(a) shows four failed retrieval examples at an altitude of 150 m. When the altitude increases to 300 m, as shown in Figure 13(b), loc1 and loc2 are successfully retrieved, whereas loc3 and loc4 still fail. The improvements for loc1 and loc2 can be attributed to the broader field of view at higher altitude, which allows the model to exploit salient visual keypoints (e.g., bridges and building rooftops) to encode structural semantics consistent with the satellite view. In contrast, loc3 and loc4 remain unmatched despite containing discriminative information. For loc3, cross-view attribute discrepancies, such as variations in rooftop color and seasonal landscape changes, impede reliable correspondence. For loc4, although prominent cues like white building facades are present, the model is misled by competing visual regions (e.g., playgrounds), resulting in retrieval failure.

## E. Discussion

This work presents InfoGeo, a cross-view object-centric learning framework for UAV-based CVGL, along with an information-theoretic perspective that explains its effectiveness. From the standpoint of Information Bottleneck theory, we reformulate cross-view object-centric learning as the joint optimization of two complementary objectives: suppressing

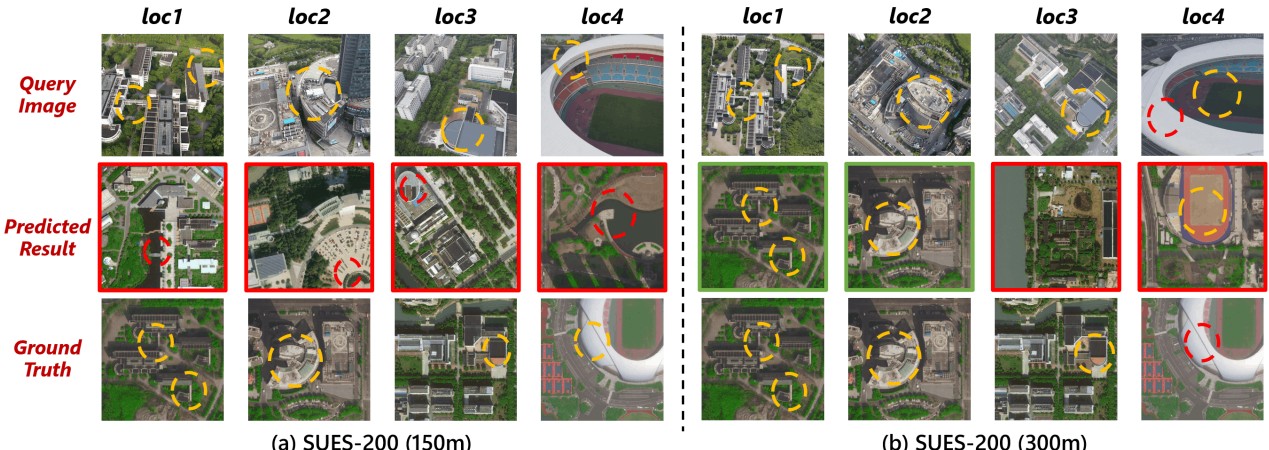

*Figure 13.* The retrieval results of the failure case in University-1652→SUES-200@(150 m and 300 m). The predicted result is the top-1 retrieval results. Dash circles denote the key visual clues (Red lines denote the wrong-matched patterns, while Yellow lines are the discriminative objects).

view-specific noise and preserving view-invariant semantic information. The proposed cross-view adaptive concept selection module filters appearance-dependent noise by leveraging collaborative cross-view cues, while the concept structural relational reasoning module captures inter-object structural relations to maximize cross-view mutual information. In addition, channel-wise FiLM blocks integrate object-centric perceptual cues into global representations, and Relational Distillation further improves performance while avoiding the inference-time overhead introduced by object-centric learning modules. Extensive experiments on multiple UAV benchmarks demonstrate that this design yields robust and generalizable representations under significant viewpoint, altitude, and weather conditions, validating the effectiveness of InfoGeo for real-world cross-view geo-localization.

Some interesting phenomena are observed in our experiments. (1) On the SUES-200 dataset, InfoGeo demonstrates particularly strong performance at 150 m, where other methods typically underperform. However, at 300 m, the model does not achieve substantial gains. Quantitative results in Table 9 and visualization in Figure 13 analyses suggest that the increased altitude and broader field of view introduce a larger number of objects, which complicates the identification of the most discriminative concepts. (2) The OCL framework exhibits significant improvements in robustness under multi-weather conditions and in qualitative visualizations (see Figure 12). This indicates that extracting view-invariant concepts while suppressing view-specific noise effectively enhances both model robustness and cross-domain generalization. (3) In integrating OCL with the CVGL task, two observations emerge. First, during training, the high-level reconstruction weight $\lambda_1$ should remain moderate; excessive values cause slot collapse, undermining object diversity. Second, when fusing Object-Centric and original representations, the fusion parameter $\alpha$ exhibits a "high-middle-high" trend: optimal performance occurs when one representation dominates, although reasonable performance can still be obtained without fusion. These findings motivate future work to further investigate the operational scope and deployment scenarios.

