# OpenReview forum: "InfoGeo: Information-Theoretic Object-Centric Learning for Cross-View Generalizable UAV Geo-Localization"
_ICML.cc/2026/Conference — ICML 2026 regular_

### Official Review · Reviewer_gNLD · 2026-03-02

**Soundness:** 3
**Presentation:** 3
**Significance:** 3
**Originality:** 3
**Overall Recommendation:** 4
**Confidence:** 3

**Summary:**

The paper tackles the problem of cross-view geo-localization (CVGL), specifically aiming to match UAV imagery with satellite views in the presence of severe domain shifts, such as varying regional textures and weather conditions. The authors propose InfoGeo, an information-theoretic framework based on Object-Centric Learning (OCL), designed to enhance robustness. The approach formulates the matching optimization as an Information Bottleneck process, which maximizes view-invariant information and minimizes view-specific noisy signals. The method is evaluated on four UAV benchmarks and demonstrates superior generalization capabilities.

**Compliance With Llm Reviewing Policy:**

Affirmed.

**Key Questions For Authors:**

1. Table 6 indicates a sharp drop in performance when the number of retained eigenvectors in the CSRR module ($r$) is increased from 4 to 8. Why does a slightly larger spectral subspace collapse discriminative local features so aggressively, and does $r=4$ universally hold as the optimal bottleneck across all datasets regardless of scene complexity?
2. Given the high sensitivity of the model to the auxiliary hyperparameters $\lambda_1$ and $\lambda_2$, how were these values tuned for the zero-shot cross-dataset transfer scenarios? In a real-world deployment, would a UAV need continuous retuning of these parameters based on its target altitude or environment?

**Limitations:**

yes

**Strengths And Weaknesses:**

Strengths:
1. The manuscript elegantly frames the CVGL problem using the Information Bottleneck (IB) theory to mathematically separate view-invariant structural semantics from task-irrelevant nuisances.
2. The Cross-View Adaptive Concept Selection (CACS) module dynamically reweights concept-level decoding attention maps based on their cross-view relevance. This introduces a highly novel mechanism for suppressing environmental artifacts directly within the OCL paradigm.
3. The framework is evaluated across four diverse UAV benchmarks (University-1652, SUES-200, DenseUAV, GTA-V), demonstrating consistent state-of-the-art results. This broad evaluation proves the high practical impact and effectiveness of the proposed method.

Weaknesses:
1. he overall optimization objective relies on manually tuning multiple auxiliary parameters ($\lambda_{1}$, $\lambda_{2}$, $\lambda_{3}$) alongside the primary InfoNCE alignment loss. This increases the training complexity and reduces the method's out-of-the-box utility.
2. In complex datasets like GTA-V, discriminative cues rely heavily on higher-level structural information (e.g., road topology) rather than the precise, fine-grained object-level alignments InfoGeo optimizes. This questions the fundamental premise of fine-grained OCL for highly varied landscape topographies.

---

> ### Author Rebuttal · Authors · 2026-03-31
>
> Thank you for taking the time to review our work and for the valuable feedback. Below, we summarize the key responses to your concerns.
> > W1/Q2. Auxiliary parameters in objectives
> - Previous works [1, 2] have explored the effectiveness of end-to-end optimization of OCL modules with downstream tasks.  Although these auxiliary objectives increase training complexity, they play important roles in significantly improving generalization ability.  As illustrated in Tab.1 and 2, the OCL modules boost the performance of CVGL across all scenarios.
> - We further conduct cross-dataset experiments with five seeds on **R1↑** to evaluate the robustness of $\lambda_1$ and $\lambda_2$ in the table below (*the average results of different altitude settings in SUES-200 are reported*). These results show that InfoGeo consistently achieves clear performance gains over prior methods and remains stable when both hyperparameters are set within the range of 0.01-0.075. Therefore, it validates the robustness and practical applicability of the parameter settings, indicating that they can be directly optimized in the source dataset without additional tuning for zero-shot transfer scenarios.
> |$\lambda_1$|University→SUES|DenseUAV→SUES|$\lambda_2$|University→SUES|DenseUAV→SUES|
> |:---:|:---:|:---:|:---:|:---:|:---:|
> |0.010|93.30±0.19|90.95±0.25|0.010|91.59±0.18|90.01±0.23|
> |0.025|94.28±0.21|92.13±0.20|0.025|94.89±0.19|90.50±0.21|
> |**0.050**|**95.03±0.16**|**93.45±0.22**|**0.050**|**95.03±0.16**|**93.45±0.22**|
> |0.075|93.65±0.18|92.00±0.21|0.075|94.75±0.15|92.53±0.23|
> |0.100|90.11±0.20|88.86±0.25|0.100|90.32±0.17|91.80±0.24|
>
> > W2. Effectiveness in GTA-V
>
> In fact, the GTA-V dataset contains diverse scenes, including large-scale regions (e.g., deserts) and dense-object regions (e.g., ports, urban areas).
> - In large-scale regions, the discriminative cues rely on high-level structural information through the spatial arrangement of semantic components. And InfoGeo can model the relations among these components through Object-Centric Learning.
> - The dense regions are shown in Fig. 3 at the link (https://anonymous.4open.science/r/InfoGeo-F230/README.md). The fine-grained objects in GTA-V can be clearly disentangled as individual concepts, and our method effectively captures view-consistent semantic cues through object-level reasoning and exploits spatial relationships to improve generalization. This is also consistent with the quantitative results shown in Tab. 2, where InfoGeo achieves SOTA performance on GTA-V.
>
> > Q1. Sensitivity of $r$ in $L_{struct}$
>
> The performance degradation observed at a larger spectral subspace ($r=8$) can be explained by the spectral filtering effect. Studies [3, 4] show that the low-rank constraint plays a key role in filtering out noisy components, thereby preserving the dominant cluster structure in the graph Laplacian. Since the slot number $K$ is fixed to 16, setting $r=8$ may corrupt reliable semantic relationships. As a result, more view-specific or task-irrelevant components are preserved, leading to the encoding of noisy or redundant relations.
>
> The quantitative experiments with five random seeds are shown in the table below and evaluated using the average **R1↑** performance. The generalization performance initially improves as $r$ increases and reaches its peak when $r=4$, then declines markedly as $r$ becomes larger. These results validate that $r=4$ is the optimal value for $L_{struct}$ across various scenarios regardless of scene complexity.
> |r|University→SUES-200|DenseUAV→SUES-200|GTA-V (Cross-Area)|
> |:-:|:--------------------:|:------------------:|:------------------:|
> |2|92.58±0.20|91.74±0.27|57.32±0.30|
> |**4**|**95.03±0.16**|**93.45±0.22**|**57.83±0.27**|
> |6|93.75±0.18|92.65±0.24|55.65±0.31|
> |8|89.90±0.22|90.47±0.22|53.73±0.29|
>
> ---
> References:
> 1. Tang L, Yuan Y, Chen C, et al. Ocrt: Boosting foundation models in the open world with object-concept-relation triad[C]//Proceedings of the IEEE/CVF Conference on Computer Vision and Pattern Recognition. 2025: 25422-25433.
> 2. Wu, Ziyi, et al. "Slotdiffusion: Object-centric generative modeling with diffusion models." Advances in Neural Information Processing Systems 36 (2023): 50932-50958.
> 3. Nie F, Wang X, Jordan M, et al. The constrained laplacian rank algorithm for graph-based clustering[C]//Proceedings of the AAAI conference on artificial intelligence. 2016, 30(1).
> 4. Kumar S, Ying J, de Miranda Cardoso J V, et al. Structured graph learning via Laplacian spectral constraints[J]. Advances in neural information processing systems, 2019, 32.

---

> > ### Author Rebuttal · Reviewer_gNLD · 2026-04-02
> >
> > I appreciate your thoughtful and detailed replies to my questions and comments. You have addressed all of my concerns satisfactorily, and I have no further queries.

---

> > > ### Author Response · Authors · 2026-04-02
> > >
> > > Thank you for your thoughtful consideration and for recognizing the value our work may bring to the community. We appreciate your assessment and support.

---

### Official Review · Reviewer_pXqz · 2026-03-11

**Soundness:** 3
**Presentation:** 2
**Significance:** 2
**Originality:** 3
**Overall Recommendation:** 4
**Confidence:** 4

**Summary:**

This paper integrates Information Bottleneck theory with object-centric learning, converting the optimization objective of maximizing structural consistency mutual information and minimizing view-specific noise conditional mutual information into trainable loss function. The information loss established from the CACS loss and orthogonal Procrustes loss provides solid information-theoretic support for feature learning in UAV cross-view matching. For UAV cross-view geo-localization tasks, the InfoGeo framework composed of CACS, CSRR and OCVA is introduced to address the issues of insufficient performance and limited generalization in complex environments, enabling explicit disentanglement between view-invariant information and view-specific noise. Specifically, CACS explicitly suppresses view-unique noise and extracts view-invariant information; CSRR mitigates implicit noise arising from spatial view discrepancies and enhances cross-view structural consistency; OCVA integrates denoised view-invariant information with global features to generate high-quality global representations, which effectively alleviates visual clutter and domain shift in complex scenes.Experiments on multiple UAV benchmark datasets show that InfoGeo achieves SOTA performance against SOTA trackers, with stronger robustness in cross-region and multi-weather challenging scenarios.

**Compliance With Llm Reviewing Policy:**

Affirmed.

**Key Questions For Authors:**

1. Whether the concept selection of CACS acts as an effective prerequisite for CSRR, and whether L_CSRR can still bring performance gains when L_cacs is excluded.
2. The relationship between the weight variations of the CACS loss and the effectiveness of concept selection should be visually analyzed. It is necessary to verify whether the visualized decoder attention maps, concept selection results, and localization results under different values of W_cv align with the conclusions presented in the paper.
3. It is necessary to check whether hyperparameter α in OCVA and hyperparameter r in CSRR are overly sensitive. When r = 8, the performance is worse than that of w/o CSRR and infoNCE; is there a more reasonable explanation for this phenomenon? Besides, why does the performance change so sharply when α = 0.8?
(Inherent analysis or relevant supporting data will help improve the evaluation of the generalization and soundness of the proposed method.)
4. Why is the evaluation of the ablation experiments only conducted on a subset of the test set, and why is the 150 m subset specifically chosen?

**Limitations:**

yes

**Strengths And Weaknesses:**

Strengths:
1. Based on Information Bottleneck theory, this paper establishes a connection between the theory and the UAV cross-view geo-localization task. It puts forward the maximization of structural consistency information and the minimization of view-specific noise, and converts these mutual information objectives into trainable loss constraints, offering definite theoretical support for cross-view feature learning.
2. The formula derivations, symbol definitions and operational logic of each module show no obvious defects and inconsistencies.
Weaknesses:
1. UAV cross-view geo-localization demands very high real-time performance. The lightweight InfoGeo∗ model in this paper has a computational cost of 91.70 GFLOPs, which is similar to the baseline CVcities. This makes it’s hard to satisfy the real-time needs of UAV applications. In addition, the ablation experiments show that the proposed method is overly sensitive to hyperparameters, and there are concerns about its generalization ability.
2. The paper’s qualitative analysis only uses PCA visualization and concept affinity maps, and only shows successful examples of the proposed model. It doesn't‌ give visual comparisons with baseline models under the same experimental settings, so the model’s advantages cannot be shown clearly.
3. The physical meaning of some key formulas is only explained at a mathematical level. For example, the paper simply describes the weight constraint formula (9) for the CACS loss as “penalizing ambiguous weights and increasing variance”, but doesn't explain how weight changes directly affect concept selection.

---

> ### Author Rebuttal · Authors · 2026-03-31
>
> We appreciate the reviewer’s valuable suggestions.  Below are our point-by-point responses.
> > W1. Real-time Performance
>
> In fact, CVGL is a retrieval task rather than a low-level control task like obstacle avoidance, and thus does not require high-frequency prediction. Furthermore, InfoGeo* achieves 117 FPS (448×448), which is sufficient for real-time navigation at typical flight. For deployment, we have two solutions to reduce the actual costs: (1) Satellite features can be computed offline since they are fixed, and online computation only encodes UAV images. (2) We can follow [1] to distill knowledge from backbone into a lightweight model (e.g., MobileViT) via *RD*. In addition, recent advances [2, 3] in VLA tasks have already deployed VLMs on UAV and robotic edge devices.
>
> > W2. Limitations on Visualization
>
> We clarify that the main text primarily presents successful examples, whereas failure cases are analyzed in Fig.12 (Appendix). And more qualitative analyses are available at https://anonymous.4open.science/r/InfoGeo-F230/README.md.
>
> The visualizations of Baseline† are shown in Fig.1 (see the link). In Fog-Rain, the PCA maps of Baseline† blend the circular platform with the flowerbed, indicating feature collapse under complex variations, whereas InfoGeo can capture more distinct features. For retrieval, InfoGeo shows a false result at 150m due to object-level misalignment, but achieves more accurate localization than Baseline† at 300m. These results highlight the strong robustness and generalization of InfoGeo.
>
> > W3. More Analysis of Key Formulas
>
> Eq.9 regularizes $W_{cv}$ to realize view-specific noise compression. The first term penalizes the expectation of $W_{cv}$ tending toward 0.5, thereby encouraging them to converge to 0 or 1. This promotes emphasis on view-consistent concepts. Meanwhile, the variance term encourages more diverse concept selection. Eq.13 enforces cross-view concept structural consistency and mitigates cross-view ambiguity by preserving intrinsic and view-invariant scene structure.
>
> From a higher-level view, Eq.18 combined with Eq.9 and Eq.13 makes OCL more suitable for our task by information-theoretic principles. The details are discussed in Appendix B.2 and will be clarified in the revision.
>
> > Q1. Relationship between CACS & CSRR
>
> The two modules are theoretically complementary: $L_{struct}$ filters out task-irrelevant noise, while $L_{struct}$ enforces cross-view concept consistency. CSRR becomes more effective when combined with $L_{cacs}$, as it serves as a prerequisite for reliable concept alignment. We evaluate the performance (**R1↑**) of CSRR w/o $L_{cacs}$, which shows that $L_{struct}$ alone still brings consistent gains. However, the improvement is smaller than the full model, confirming that CACS enhances CSRR by providing cleaner and more informative concepts.
> |$L_{cacs}$|$L_{struct}$|University→SUES|DenseUAV→SUES|GTA-V (Cross-Area)|
> |:---:|:---:|:---:|:---:|:---:|
> |❌|✅|93.97|92.40|57.02|
> |✅|✅|95.07|93.41|57.90|
>
> > Q2. Visualization of $W_{cv}$
>
> As shown in Fig.2 at the link, the distribution of $W_{cv}$ becomes more selective after introducing $L_{cacs}$, assigning larger weights to view-consistent concepts and lower weights to less reliable ones. This is also supported by PCA maps and cross-view concept affinities, indicating that the retained concepts are semantically aligned across views.
>
> > Q3. Hyperparameter Sensitivity
>
> For $r$, the theoretical details and experimental results can be found in the response to Reviewer gNLD (Q1).
>
> Regarding $\alpha$, it balances the original branch (global features) and the OCL branch (view-consistent & fine-grained features). More detailed effects of $\alpha$ are shown below. It exhibits a *high-middle-high* trend, since the fused features are processed by a non-linear aggregator. However, InfoGeo still maintains strong performance when $\alpha$ lies in 0.75-0.85, suggesting the setting remains practically applicable rather than overly sensitive.
> |$\alpha$|University→SUES|DenseUAV→SUES|GTA-V (Cross-Area)|
> |:---:|:---:|:---:|:---:|
> |0.55|92.18±0.18|91.16±0.19|57.17±0.30|
> |0.65|91.26±0.21|90.99±0.23|56.95±0.29|
> |0.75|93.72±0.20|92.40±0.21|57.34±0.25|
> |**0.80**|**95.03±0.16**|**93.45±0.22**|**57.83±0.27**|
> |0.85|94.89±0.18|93.23±0.20|55.75±0.31|
>
> > Q4. 150m subset
>
> In the main text, we report the 150m subset, as it serves as a representative setting (see the response to Reviewer nBKG W1.1). More results can be found in Tab.11.
>
> ---
> References:
> 1. Sun, Jian, et al. "MobileGeo: Exploring Hierarchical Knowledge Distillation for Resource-Efficient Cross-view Drone Geo-Localization." arXiv preprint arXiv:2510.22582 (2025).
> 2. Xin Z, et al. AgentVLN: Towards Agentic Vision-and-Language Navigation[J]. arXiv preprint arXiv:2603.17670,2026.
> 3. Zheng G, et al. OnFly: Onboard Zero-Shot Aerial Vision-Language Navigation toward Safety and Efficiency[J]. arXiv preprint arXiv:2603.10682,2026.

---

### Official Review · Reviewer_nBKG · 2026-03-13

**Soundness:** 3
**Presentation:** 4
**Significance:** 3
**Originality:** 4
**Overall Recommendation:** 4
**Confidence:** 4

**Summary:**

This paper aims to generalize Cross-view Geo-Localization (CVGL), which searches for the same location across different viewpoints using UAV (drone) images as queries and satellite images as galleries, even under domain shifts such as unknown regions and unknown weather conditions. Many existing CVGL methods rely on global feature alignment (primarily contrast learning), which suffers from degraded performance due to regional texture variations, weather differences, and clutter caused by high-density objects in the UAV viewpoint. Our proposed method, InfoGeo, introduces the Object-Centric Learning (OCL) framework to UAV-CVGL. It formalizes the design principle as the Information Bottleneck principle, which extracts “structurally and geometrically stable information” shared across views while suppressing ``view-specific noise'' such as weather, occlusions, and moving objects. Experiments evaluated cross-domain transfer and multi-weather transfer on SUES using four benchmarks: University-1652, SUES-200, DenseUAV, and GTA-V. Performance was compared with existing methods using R@K and AP metrics. The study also presents ablation studies, computational complexity comparisons, visualizations, and failure case analysis.

**Compliance With Llm Reviewing Policy:**

Affirmed.

**Final Justification:**

The rebuttal from the authors has addressed my concerns, keeping the score.

**Key Questions For Authors:**

- Regarding the central assumptions of the theoretical framework as mentioned in Weakness above, how do you account for the impact when these assumptions are violated by actual data? Could you provide quantitative and qualitative evidence (such as additional experiments or diagnostic indicators) demonstrating the validity of these assumptions? If evidence of robustness against assumption violations is presented, it would enhance the persuasiveness of the theoretical claims and improve the soundness evaluation.

**Limitations:**

Yes

**Strengths And Weaknesses:**

Strengths:
- The paper maps the objectives of IB (task-related information retention and noise compression) onto CVGL's cross-view learning. It presents the reasoning—through theorems (IB principle, cross-view mutual information inequality) and propositions—for why maximizing cross-view mutual information aids in location identification, both in the main text and appendix. Particularly noteworthy within the paper is the derivation of inequalities based on Markov chains and the Data Processing Inequality, further demonstrating that concept-level mutual information (I(Uq;Ug)) serves as a lower bound for mutual information between latent representations.
- Experiments span multiple benchmarks, covering transition scenarios (multiple), multi-weather, altitude-specific evaluations, ablation studies of key components/losses/hyperparameters, computational complexity comparisons, visualizations, and failure case analysis, aiming to verify the contribution of each proposed element.

Weaknesses:
- Minor typos (e.g., mianly, whle, baselinet) are occasionally found in the text. Please recheck the entire paper.
- The text and appendix mention phenomena like smaller improvements around 300m altitude and performance degradation when introducing RD, indicating limitations in applicability.
- The core of the theory relies on assumptions of conditional independence (e.g., $\hat{Z}q \perp \hat{Z}g|Y)$) and assumptions about information processing chains such as $Y \to \hat{Z}g \to Zq \to \hat{Z}q$. The extent to which these assumptions hold in real data (especially in UAV scenarios with strong clutter) requires verification and and discussion of how well these assumptions hold in real data (especially in clutter-heavy UAV scenarios) appears limited within the text (at least, while the assumptions themselves are explicitly stated, the impact of violating them is not systematically evaluated).

---

> ### Author Rebuttal · Authors · 2026-03-30
>
> We appreciate the time and effort you have dedicated to our manuscript. Below, we summarize the key responses to your concerns.
>
> > W1. Minor Typos
>
> Thank you for the careful review. We will correct all the typographical errors (e.g., the caption of Figure 4 should be **'while dashed circles ...'**) in the revised manuscript.
>
> > W2. Applicability across altitudes and effectiveness of *RD*
>
> **W2.1 Applicability across Altitudes**
>
> As shown in Fig. 12 (Appendix), UAV images are visually closer to satellite views at 300m. All methods benefit from the smaller cross-view gap, making the relative gain of InfoGeo less pronounced (a similar trend is also observed in the Baseline).
>
> In contrast, images at 150m differ more substantially from satellite views, resulting in more significant cross-view ambiguity. The superior performance of InfoGeo at 150m further demonstrates its effectiveness in mitigating this issue.
>
> **W2.2 Effectiveness of *RD***
>
> In our work, *RD* is introduced to improve inference efficiency by avoiding additional computational modules, leading to a parameter count nearly identical to Baseline. In Tab. 5 and 9, *RD* consistently improves performance on University-1652 and also brings gains on DenseUAV→SUES-200@(150m/200m), while performance degradation is observed on DenseUAV→SUES-200@(250m/300m).
>
> Since DenseUAV is a low-altitude dataset (80–100m), the relational knowledge distilled by *RD* may contain altitude-specific biases. Importantly, both the gains and degradations introduced by *RD* are relatively minor. This suggests that the performance improvement of InfoGeo mainly comes from the Cross-View OCL framework, while *RD* mainly serves as an auxiliary module that further supports the lightweight design of our proposed method.
>
> > W3/Q1. The theoretical assumptions
>
> This is a critical comment for our work. These assumptions in the main text (Sec.3) and Appendix B.2 hold in the idealized CVGL setting, where the two views are captured by different platforms with independent sensors and imaging conditions, and are coupled only through the shared geographic information $Y$.
>
> To empirically verify the assumptions, we further conduct Mutual Information (MI) analyses on training sets in the table below. $I(Z_q, Z_g)$ denotes the overall statistical dependence between query and gallery features, reflecting nontrivial global cross-view dependence. However, both the conditional mutual information $I(Z_q, Z_g \mid Y)$ (using pretrained DINOv2) and $I(\hat{Z}_q, \hat{Z}_g \mid Y)$ (using InfoGeo) are near zero in all benchmarks, indicating no statistically significant dependence remains across two views. These results validate that the conditional independence assumption is reasonable in practice and provide strong support for the Markov chain formulation, i.e., $Y \to \hat{Z}_g \to Z_q \to \hat{Z}_q$.
> | Metric | University-1652 | DenseUAV | GTA-V |
> |:---|:---:|:---:|:---:|
> | $I(Z\_q, Z\_g)$ | 0.3472 | 0.0797 | 0.1540 |
> | $I(Z\_q, Z\_g \mid Y)$ | 0.0000 | 0.0000 | 0.0000 |
> | $I(\hat{Z}\_q, \hat{Z}\_g \mid Y)$ | 0.0000 | 0.0000 | 0.0000 |
>
> Nevertheless, these assumptions may not strictly hold in all real-world applications. For example, some scenes may lack salient cross-view discriminative objects. In such cases, the model can still capture view-consistent textured regions (e.g., characteristic vegetation distributions or road orientations) to construct concepts for object-centric learning.
>
> GTA-V is a representative example of the clutter-heavy scenarios, as it contains many complex scenes in which cross-view correspondences are often partially matched rather than one-to-one matches. As shown by the quantitative and qualitative results in Tab. 2 and Fig. 10, InfoGeo still achieves strong performance on GTA-V, suggesting that these assumptions are better viewed as first-order analytical approximations that motivate our training objective.

---

> > ### Author Rebuttal · Reviewer_nBKG · 2026-04-04
> >
> > The rebuttal clarifies that the minor typos will be corrected, so that concern has been resolved. Regarding altitude, the explanation that UAV images at 300m become visually closer to satellite views, so that all methods benefit and the relative gain of InfoGeo appears smaller, while the advantage of InfoGeo becomes more evident at 150m where the cross-view gap is larger, largely addresses my concern about how to interpret these results. My concern about RD has also been substantially alleviated: the rebuttal makes clear that RD is introduced mainly as an auxiliary component for improving inference efficiency, that it can lead to both gains and degradations, that its behavior may reflect altitude-specific bias because DenseUAV is a low-altitude dataset, and that the magnitude of these effects is relatively small.
> >
> > However, for the theoretical assumptions (in particular the conditional independence assumption and the resulting Markov chain formulation), the additional MI analysis substantially improves my confidence by showing near-zero conditional MI. That said, the strength of this evidence is still difficult to assess from the rebuttal alone, because it remains unclear how the MI and conditional MI were estimated in practice, including the choice of estimator, the treatment of the data, and how conditioning on $Y$ was implemented. It is also unclear whether 0.0000 reflects rounding or an effectively exact zero.

---

> > > ### Author Response · Authors · 2026-04-04
> > >
> > > We sincerely thank the reviewer for the constructive discussion and for acknowledging that our previous responses have resolved some of your concerns.
> > >
> > > We briefly clarify the remaining theoretical concern. These assumptions function as an **information-theoretic principle** and an **analytical approximation** to motivate the core training objective. Specifically, this approximation encourages the model to filter out view-specific nuisances and extract invariant object-centric concepts for cross-view matching. In complex real-world scenarios, our method does not enforce strict one-to-one isolated object extraction. Instead, when the idealized assumptions become less accurate (e.g., in clutter-heavy datasets such as GTA-V), the Information Bottleneck objective can still utilize view-consistent patterns, such as textured regions, vegetation, or road topologies, to construct object-centric concepts. The strong empirical gains on GTA-V (Table 2) further suggest that **InfoGeo remains robust in practice, even when these theoretical approximations are imperfect.**
> > >
> > > For the empirical MI analyses in our rebuttal, they should be interpreted as **diagnostic evidence** rather than a formal verification. The MI was estimated using a standard non-parametric kNN estimator ($k=3$). Meanwhile, the conditional MI was approximated by grouping samples according to the shared geographic identity \(Y\), using the same estimator to calculate the group-level mutual information (group size $\geq$ 3), and averaging the resulting values across groups. For data treatment, a zero-variance test was applied before MI estimation to filter features. As you astutely noted, the reported **"0.0000" is not an exact mathematical zero, but a rounded value below the current reporting precision ($<10^{-4}$)**. We agree that this point should be stated more carefully. And the details on the estimator, data treatment, and conditioning procedure for the empirical analyses will be clarified in the revised appendix.

---

### Decision · Program_Chairs · 2026-04-30

**Decision:**

Accept (regular)

**Comment:**

This paper aims to generalize Cross-view Geo-Localization (CVGL) by integrating Information Bottleneck theory with object-centric learning. It converts the optimization objective of maximizing structural consistency mutual information and minimizing view-specific noise conditional mutual information into the loss. The novelty is appreciated by all reviewers and the experimental evaluation is comprehensive, which supports the proposed method.

After the two-round rebuttal, the recommendations from all reviewers reach a consensus. I'd like to recommend it for acceptance.